# Controlling the pandemic during the SARS-CoV-2 vaccination rollout

João Viana [1,2], Christiaan H. van Dorp [3], Ana Nunes [2,4], Manuel C. Gomes [2], Michiel van Boven [1], Mirjam E. Kretzschmar [1], Marc Veldhoen [5] & Ganna Rozhnova [1,4 ✉]

There is a consensus that mass vaccination against SARS-CoV-2 will ultimately end the COVID-19 pandemic. However, it is not clear when and which control measures can be relaxed during the rollout of vaccination programmes. We investigate relaxation scenarios using an age-structured transmission model that has been fitted to age-specific seroprevalence data, hospital admissions, and projected vaccination coverage for Portugal. Our analyses suggest that the pressing need to restart socioeconomic activities could lead to new pandemic waves, and that substantial control efforts prove necessary throughout 2021. Using knowledge on control measures introduced in 2020, we anticipate that relaxing measures completely or to the extent as in autumn 2020 could launch a wave starting in April 2021. Additional waves could be prevented altogether if measures are relaxed as in summer 2020 or in a step-wise manner throughout 2021. We discuss at which point the control of COVID-19 would be achieved for each scenario.

[1] Julius Center for Health Sciences and Primary Care, University Medical Center Utrecht, Utrecht University, Utrecht, The Netherlands. [2] Faculdade de Ciências, Universidade de Lisboa, Lisbon, Portugal. [3] Theoretical Biology and Biophysics (T-6), Los Alamos National Laboratory, Los Alamos, NM, USA. [4] BioISI—Biosystems & Integrative Sciences Institute, Faculdade de Ciências, Universidade de Lisboa, Lisbon, Portugal. [5] Instituto de Medicina Molecular João Lobo Antunes, Faculdade de Medicina, Universidade de Lisboa, Lisbon, Portugal. ✉email: g.rozhnova@umcutrecht.nl

Mass vaccination against SARS-CoV-2 started in Europe in late 2020 and early 2021[1], and brings hope that the COVID-19 pandemic can be brought to an end in 2021. Even though progress towards this goal seems to be on the right track[2], many governments in Europe continue to limit socio-economic activities to control the pandemic. Despite elaborate national vaccination schedules, it is unclear when and which control measures can be relaxed and at which point the control of the pandemic will be achieved during the vaccination pro-gramme. Understanding of how relaxation policies might affect the transmission dynamics of SARS-CoV-2 is furthermore hampered by the emergence of novel variants[3,4] that have a selective advantage, such as increased transmissibility[5–8] or the ability to reduce rapid neutralization by the host[9]. For example, the current restrictions in Europe[10] are in part caused by a more transmissible[5–8] and potentially more pathogenic[11,12] B.1.1.7 variant that originated in the UK and is quickly gaining dom-inance in other countries, including Portugal[8,13,14].

The vaccines that have been approved in Europe[15] show consistently high efficacy against severe disease, hospitalization and death in trials[16–18] and show equally high effectiveness in real-world settings[19–23]. Multiple studies are under way to establish infection-blocking properties of these vaccines. Analyses of the national vaccination programme in Israel indicate that the effectiveness of the Pfizer-BioNTech vaccine against asympto-matic SARS-CoV-2 infections could be as high as 94%[22], as announced recently by the Israel Ministry of Health, Pfizer Inc and BioNTech SE. The recent Danish cohort study on long-term care facility residents and healthcare workers further suggests that the effectiveness of the Pfizer-BioNTech vaccine using a positive PCR test as outcome measure is 64% and 90% beyond seven days of second dose in the two groups, respectively[20]. Similar results were found in a study among healthcare workers in England where the effectiveness of the Pfizer-BioNTech vaccine against symptomatic and asymptomatic infection was 86% seven days after two doses[23]. Based on the data from Israel, the effectiveness of the same vaccine against infection with SARS-CoV-2 was shown to be 51% 13–24 days after one dose[21]. Finally, in a study by Lipsitch and Kahn[24], the lower bound for the efficacy against transmission for one dose of Moderna vaccine was estimated at 61%, and could possibly be considerably higher, especially after two doses.

The consequences of relaxing control measures such as phy-sical distancing, school closure, mask-wearing, test-and-trace and isolation, will depend on several factors, including the properties of vaccines deployed in a given country, specifics of the vacci-nation schedule, and speed of vaccine rollout, but also the past epidemiology of SARS-CoV-2 that determines which fraction of the population is protected by natural infection[25,26]. All these factors are clearly country-dependent and will play a major role in how the pandemic will unfold under different relaxation scenarios[27–30] and how quickly the full control of COVID-19 will be gained in specific countries throughout 2021 and possibly beyond. To make a few distinctive examples, we recall Israel which has the highest vaccination rate worldwide so that, on average, every person has received at least one vaccine dose by mid-March 2021[1] and Manaus in Brazil, where the levels of protection by natural infection close to the theoretical herd immunity threshold were achieved prior to the start of mass vaccination[31].

An extensive body of literature addresses the challenges of real-time modeling the COVID-19 pandemic[32]. Mathematical trans-mission models robustly calibrated to available data are among the best tools available to provide input into the discussion on the response to the COVID-19 pandemic[33–43] and they will continue to play an important role in making decisions surrounding the relaxation of measures in 2021[27–30,44]. Several modeling studies provided support for the development of COVID-19 vaccines and early planning of vaccination scenarios and rollouts[45–49]. These models, however, assumed that a large proportion of the population is vaccinated instantaneously or did not focus on relaxation strategies. More recently, organized teams of modeling experts supporting decision-makers over health emergencies in China, Australia and the UK evaluated the roadmap scenarios for relaxation of control measures in these countries in light of ongoing mass vaccination[27–29,50].

The present study makes a contribution towards better understanding of when and which control measures can be relaxed as mass vaccination programmes progress in 2021. We take Portugal as a case study where good quality data for model parameterization are available but, apart from efforts of genomic surveillance[51] and a recent study on the pre-vaccination dynamics of COVID-19[52], there are few dedicated COVID-19 modeling studies for informing policymaking in this country[53]. Using an age-structured transmission model that has been fitted in a Bayesian framework to the data from various sources (age-specific hospitalizations and seroprevalence, social contact and demographic data, national vaccination plan and vaccine rollout data etc.), we investigate future pandemic trajectories under several alternative relaxation scenarios throughout 2021. Among the explored strategies are (i) lifting measures to the same extent as in summer 2020 and (ii) later on in autumn 2020, (iii) the complete lifting of measures, and (iv) combinations of (i), (ii) and (iii). We evaluate the impact of each scenario on the epidemic dynamics as quantified by projected hospital admissions, the time-dependent effective reproduction number, population immunization level due to natural infection and vaccination, and the timing of reaching control of COVID-19 in Portugal. Finally, we discuss the implications of our findings for the post-pandemic dynamics of SARS-CoV-2.

## Results

**Model calibration**. The model was fitted to age-stratified COVID-19 hospitalization data in the period from 26 February 2020 till 15 January 2021 and cross-sectional age-stratified SARS-CoV-2 seroprevalence data assessed from 21 May 2020 till 8 July 2020. The model reproduces well the age-specific hospital admissions (Fig. 1) featuring (i) the first pandemic wave (March–April 2020), (ii) relatively low epidemic activity (May–August 2020), (iii) the second pandemic wave (September-mid-December 2020), (iv) the third wave that started in mid-December 2020 and was still ongoing on 15 January 2021[54]. The estimated hospitalization rates increase with age from 0.12 (95% CrI 0.07–0.23) per year for children under 5 years of age to 14.24 (95% CrI 9.91–21.23) per year for persons above 80 years (Sup-plementary Fig. 1). In agreement with other studies[55,56], the estimated susceptibility to SARS-CoV-2 increases with age (Supplementary Fig. 2). The meaning of model parameters is given in Supplementary Tables 1 and 5, and their estimates are shown in Supplementary Figs. 1 and 2.

The model also reproduces well the age-specific and total seroprevalence in the population (Fig. 2). The estimated age-specific seroprevalence ranged between 1.77% (95% CrI 0.98–2.91%) for 1–10 years old children to 4.61% (95% CrI 3.47–5.91%) for 20–40 years old adults (Fig. 2a). The total seroprevalence steadily increased with time reaching 19.37% (95% CrI 14.82–24.57%) on 15 January 2021 (Fig. 2b).

**Time-varying contact patterns and effective reproduction number**. We estimated how age-specific contact rates in the population changed due to control measures as the pandemic

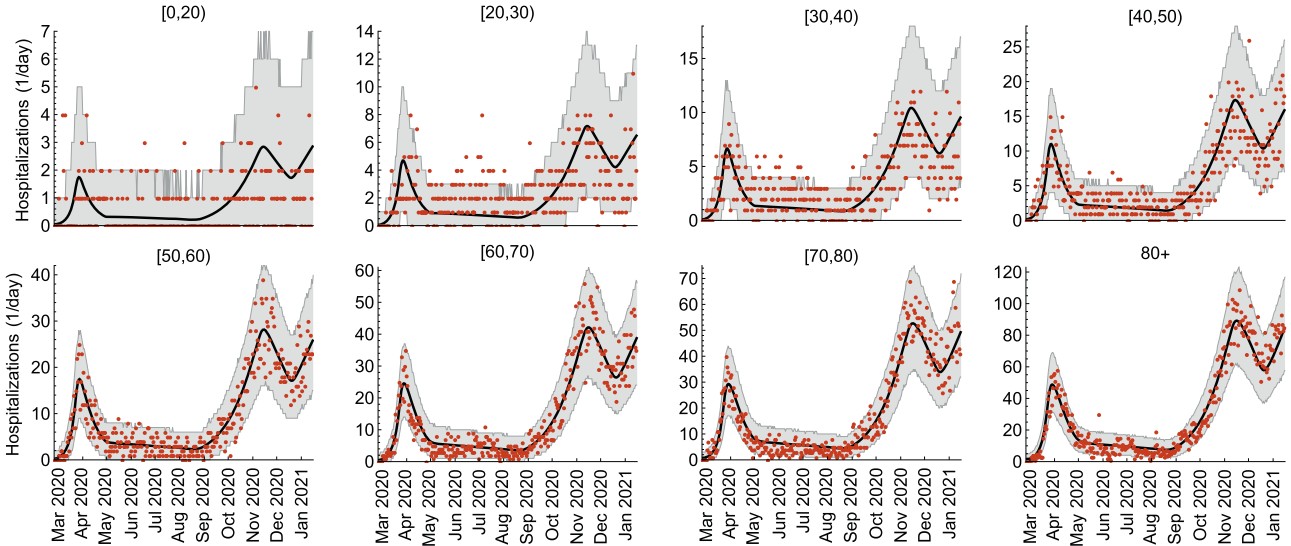

**Fig. 1 Model fit to COVID-19 hospitalizations.** The age-stratified daily hospital admission data are shown as red dots. The median trajectories estimated from the model are shown as the black lines. The gray shaded regions correspond to 95% Bayesian prediction intervals based on 2000 parameter samples from the posterior distribution. Hospital admissions were estimated for 10 age groups (see Methods). For presentation purposes, we grouped hospitalizations for ages [0,5), [5,10), [10,20) into the group of [0,20), so only 8 age groups are shown in this figure.

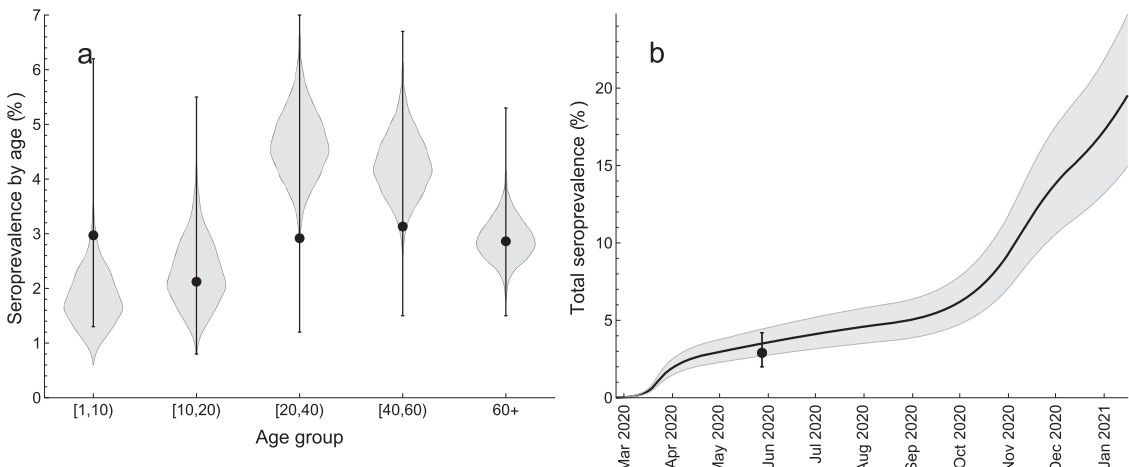

**Fig. 2 Model fit to SARS-CoV-2 seroprevalence. a** Age-specific seroprevalence. The violin shapes represent the marginal posterior distribution of the age-specific seroprevalence in the model. **b** Total seroprevalence. The black line and the gray shaded region show the median total seroprevalence and 95% credible intervals. The uncertainty in the model is based on 2000 parameter samples from the posterior distribution. The data (dots - percentage seroprevalence and error bars - 95% confidence intervals) in **a** and **b** are taken from the cross-sectional seroepidemiological survey (First National Serological Survey) conducted after the first pandemic wave[59] and supplied in the Mathematica notebook available in the GitHub repository, https://github.com/lynxgav/COVID19-vaccination[57]. The total seroprevalence refers to population older than 1 year[59].

developed. These contact rates denote the average number of transmission-relevant contacts per day a person in a given age category has with persons in other age categories. We further calculated the time-dependent effective reproduction number, $R_e(t)$, defined as the average number of secondary infections caused by one infectious individual in the population with age-specific contact patterns and age-specific seroprevalence at time $t$. $R_e(t) < 1$ signifies the control of the pandemic with possibly some of control measures in place. The full control of COVID-19 is achieved when $R_e(t) < 1$ and the contact rates in the population are restored to the pre-pandemic level.

Our findings are summarized in Fig. 3, where we show the total daily hospitalizations (Fig. 3a), the average (over all ages) number of daily contacts in the population (Fig. 3b) and $R_e(t)$ (Fig. 3c) evaluated bi-weekly in the period from 26 February 2020 till 15 January 2021. The green vertical lines indicate the estimated mid-

point transitions in the age-specific contact rates (see Methods). The pre-pandemic average number of daily contacts was 12.6. The estimated basic reproduction number (in the absence of control measures and with zero seroprevalence) was 2.20 (95% CrI 1.97–2.56). The control measures introduced during the first wave in spring 2020 reduced the number of contacts to 4.2 (95% CrI 3.3–5.0) and $R_e$ to 0.69 (95% CrI 0.64–0.75). After some of these measures were lifted, the number of contacts increased to 5.9 (95% CrI 5.1–6.6) and $R_e$ increased to almost 1 and stayed nearly constant throughout summer 2020. At the start of the second wave in autumn 2020 that followed the opening of schools and the associated changes in the contact patterns of the rest of the population, the average number of contacts further increased to about 7.6 (95% CrI 6.7–8.3) and $R_e$ to 1.24 (95% CrI 1.21–1.28). The reinforcement of measures during the second wave could only reduce $R_e$ to 0.89 (95% CrI 0.86–0.99) as

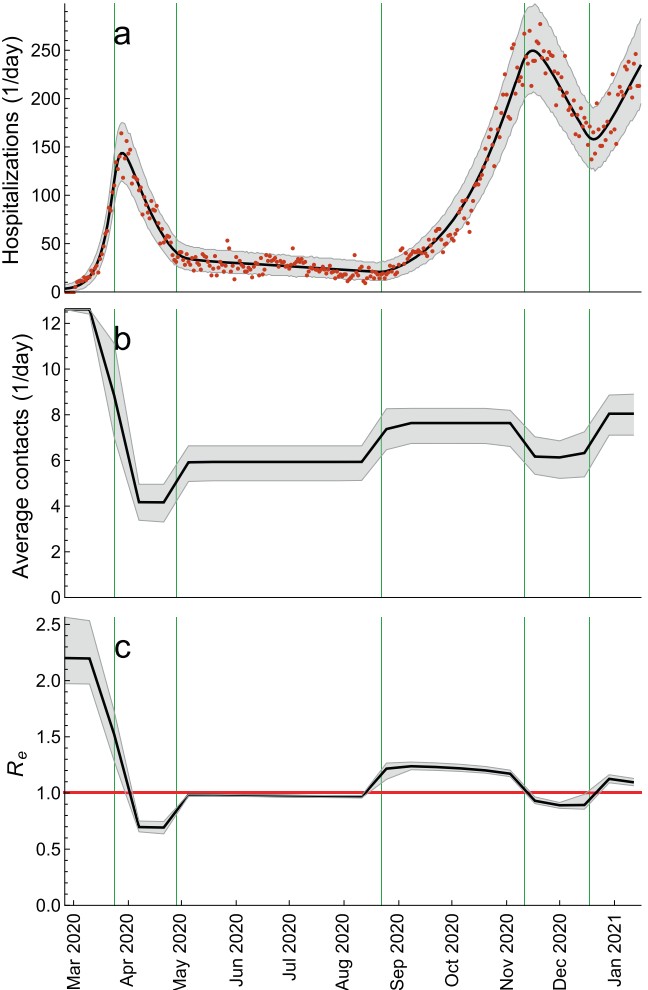

**Fig. 3 Estimated contact rate and effective reproduction number. a** Total daily hospital admissions with COVID-19. **b** Average (over all ages) number of daily contacts in the population. **c** Effective reproduction number, $R_e(t)$. The average daily contacts and $R_e$ were evaluated once every two weeks. The green vertical lines indicate the estimated mid-point transitions in the age-specific contact rates. The red horizontal line denotes $R_e = 1$. The hospitalization data are shown as red dots. The black solid lines are the median trajectories estimated from the model. The gray shaded regions correspond to 95% credible intervals.

compared to $R_e$ of 0.69 after more severe measures introduced during the first wave. Finally, the increased activity of the population around Christmas and the New Year 2021 initiated the third wave in January 2021.

**Vaccination rollout**. We implemented the rollout of vaccination against SARS-CoV-2 as set out prior to the start of the vaccination campaign by the Directorate-General of Health — a division of Portuguese Ministry of Health concerned with public health (Table 1)[57]. The mass vaccination started on 27 December 2020, is planned to proceed in three phases that will cover the whole population of Portugal by 31 December 2021. In the model results presented in the main text, we made several simplifying assumptions regarding vaccination, i.e., 1) at most 90% of each age group will be vaccinated (as supported by the survey conducted between 23 January and 5 February 2021 on the willingness to get vaccinated where the percentage of the Portuguese residents who want to get vaccinated exceeds 95%[58]) except for persons under 20 years of age (as supported by the current

guidelines on the ineligibility for vaccination of persons under 18 years of age); 2) the distributed vaccine is by BioNTech/Pfizer brand (as supported by the recent ECDC vaccination data for Portugal where 96% of vaccine doses distributed up until February 21, 2021 are by BioNTech/Pfizer); 3) vaccination is modeled as a single event that immediately confers protection equivalent to two vaccine doses; 4) we considered an infection-blocking vaccine and formulated optimistic assumptions for vaccine efficacies in reducing infection, disease and severe disease; 5) there is no waning of protection against (re-)infection after natural infection and vaccination. More details of the vaccination model are given in Methods. In the sensitivity analyses, we explored the impact on hospitalizations of timings of different relaxation steps, pessimistic assumptions for vaccine efficacies, infectivity of breakthrough cases in vaccinated persons, behavior compensation post-vaccination and the maximum age-specific coverage decreasing with age.

We used the rollout schedule (Table 1) and data (Fig. 4a) on the age distribution of morbidities among the Portuguese residents and age distribution of prioritized vaccination categories (e.g., healthcare workers, long-term care facilities staff and residents etc.) to calculate age-specific vaccination rates (number of persons in a given age group vaccinated per day) as the vaccination programme progresses (Fig. 4b; see Supplementary Fig. 3 for detailed information). The vaccination rate refers to vaccination with two vaccine doses. The maximum vaccination coverage of 90% is projected to be reached in the following order (Fig. 5a): 80+ (29 June 2021), [60,80) (20 July–23 July 2021), [50,60) (29 August 2021) and [20,50) (16 November 2021) (see Supplementary Fig. 4 for absolute numbers of vaccinated persons). The total coverage in the population will increase to 9%/38%/73% (maximum coverage) by 1 May/1 August/16 November 2021 (Fig. 4b). The vaccination rollout data based on fully vaccinated persons for Portugal[1] agree well with these projections.

**Scenarios for relaxation of control measures**. To account for the epidemiological situation in Portugal between mid-January and mid-March 2021[54], we modeled the third wave of hospitalizations that was curbed by the substantial reinforcement of measures similar to those implemented during the first wave in spring 2020. We also modeled an increase in the transmisibility of the virus due to the rapid spread of B.1.1.7. variant in Portugal. The situation in mid-March 2021 is then described by the average number of daily contacts of 4.2, $R_e$ of 0.67 and the circulating variant that is 50% more transmissible[5–7] than the original variant that was dominant in Portugal until December 2020.

Starting from this situation, we generated scenarios for relaxation of control measures as follows (Fig. 6): Scenario 1) lifting all measures so that contact rates in the population return to the pre-pandemic level (average rate of 12.6 contacts/day); Scenario 2) partial lifting of measures that increases contact rates to the level of September–October 2020 (7.6 contacts/day); Scenario 3) partial lifting of measures that increases contact rates to the level of June–August 2020 (5.9 contacts/day). In accordance with the plan of the Portuguese government to alleviate some of the current measures in spring 2021 and to make the scenarios comparable, we used the same mid-point (1 April 2021) and the same speed of transition between the contact levels (10 days).

The comparative analysis of Scenarios 1, 2, and 3 is shown in Fig. 6. The model predicts that lifting all measures (Scenario 1; Fig. 6a–d) launches a fourth wave that is significantly larger than the previous waves, resulting in 58,226 cumulative hospitalizations between 1 April 2021 and 1 January 2022 (Fig. 6a). $R_e$

**Table 1 The Portuguese vaccination plan.**

| Category | Age (years) | Vaccination period | Persons |
|---|---|---|---|
| **Phase 1** | | | **937,361** |
| Healthcare workers (HCW) | 20–65 | 27 Dec 2020–28 Feb 2021 | 199,708 |
| Long-term care facilities (LTCF) | | 01 Jan 2021–28 Feb 2021 | 148,119 |
| Residents | 65+ | | 86,982 |
| Staff | 20–65 | | 61,138 |
| Risk Group 1 | 50+ | 01 Feb 2021–30 Apr 2021 | 513,634 |
| Cardiac insufficiency | | | 207,571 |
| Coronary heart disease | | | 169,265 |
| Renal insufficiency | | | 8201 |
| Chronic obstructive pulmonary disease (COPD) | | | 128,597 |
| First response professionals (FRP) (firemen, police, military etc.) | 20−65 | 01 Feb 2021−30 Apr 2021 | 75,900 |
| **Phase 2** | | | **3,333,191** |
| Persons with or without morbidities unvaccinated before[a] | 65+ | 01 May 2021–31 Jul 2021 | 1,873,349 |
| Risk Group 2 | 50–65 | 01 May 2021–31 Jul 2021 | 1,459,842 |
| Diabetes | | | 222,864 |
| Neoplasm | | | 114,246 |
| Hepatic insufficiency | | | 93,004 |
| Chronic kidney disease | | | 4222 |
| Obesity | | | 392,959 |
| High blood pressure | | | 632,547 |
| **Phase 3** | | | **6,529,448** |
| Remaining persons (excluding children)[b] | 20–65 | 01 Aug 2021–31 Dec 2021 | 6,529,448 |
| Total[a] | | | 10,800,000 |

[a]The Portuguese vaccination plan as set out prior to the start of the vaccination campaign assumes that all persons in the population will be vaccinated with a two-dose vaccine schedule. In the model, the maximum vaccination coverage in any age group is 90%.
[b]According to the current guidelines, persons under 18 years old are not eligible for vaccination. In the model, we assumed that the age group of 0–20 years old is not vaccinated.

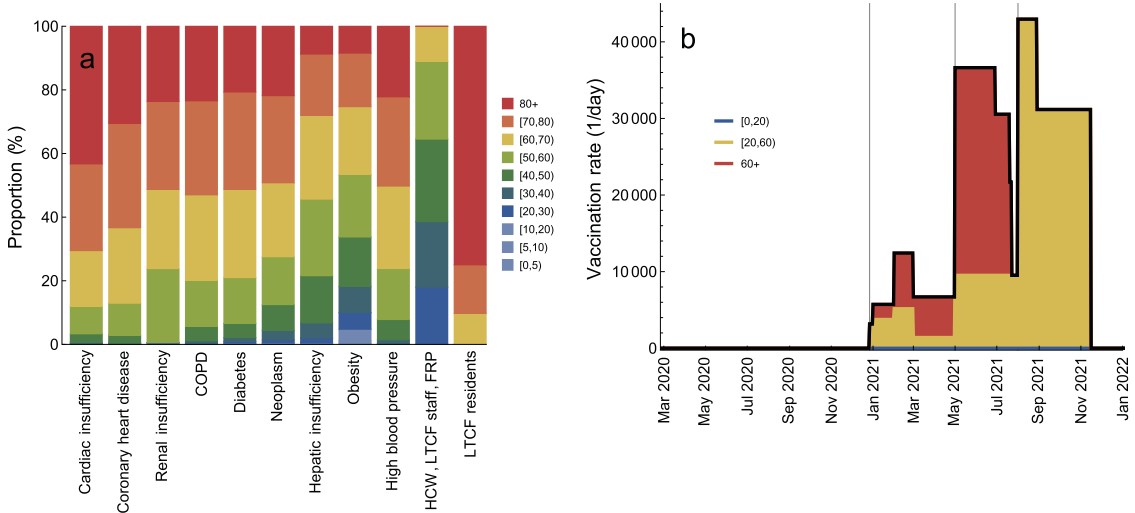

**Fig. 4 Vaccination rollout schedule. a** Age distribution of vaccination categories. **b** Total vaccination rate (number of persons vaccinated per day, black line) and proportions of vaccination rate attributable to ages [0,20) (blue), [20,60) (yellow) and 60+ (red). The gray vertical lines in **b** indicate the starting dates for different vaccination phases (Table 1). The age-specific vaccination rates are given in Supplementary Fig. 3.

increases sharply from 0.67 on 23 March 2021 to 2.03 two weeks later (Fig. 6c) which is very close to the basic reproduction number of 2.20 at the start of the pandemic. The full control over COVID-19 is reached on 18 May 2021 when $R_e$ drops below 1 and the contact rates are the pre-pandemic level (Fig. 6b). At this threshold, 60% of the population acquired protection after natural infection and only 10% are protected after vaccination (Fig. 6d). Relaxing measures according to Scenario 2 (Fig. 6e–h) initiates a new pandemic wave too, albeit smaller in magnitude than Scenario 1 (8975 hospitalizations between 1 April 2021 and 1 January 2022; Fig. 6e). In this case, $R_e$ becomes smaller than 1 on 29 June 2021 (Fig. 6g) but the measures have to be kept in place (Fig. 6f) to control the spread. The increase of contact rates to the level of June–August 2020 (Scenario 3; Fig. 6i–l), however, does not lead to a significant rise in hospitalizations (1450 hospitalizations between 1 April and 1 January 2022; Fig. 6i) because $R_e$

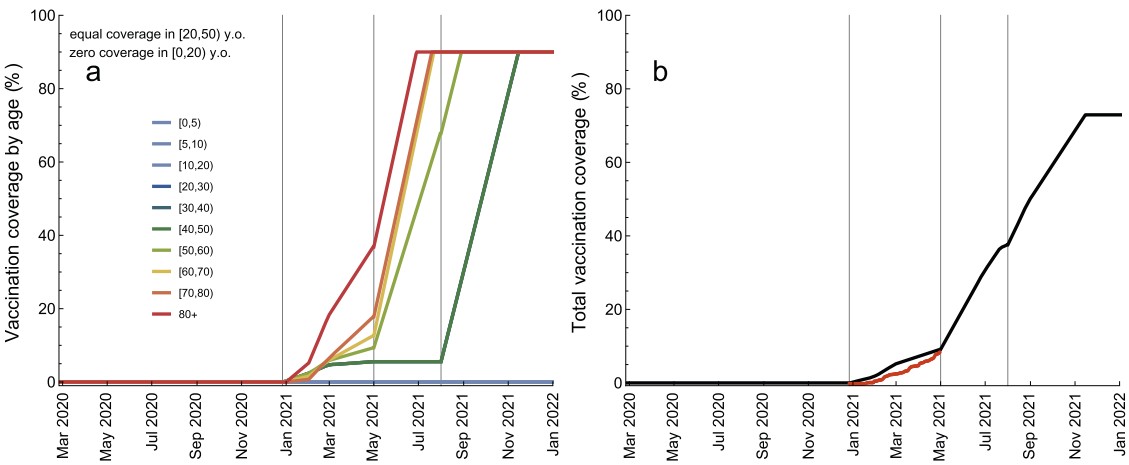

**Fig. 5 Vaccination coverage during the vaccination rollout. a** Age-specific coverage (percentage of vaccinated persons per age group). **b** Total vaccination coverage (percentage of vaccinated persons in the population). The gray vertical lines indicate the starting dates for different vaccination phases (Table 1). The coverages for ages [20,30), [30,40), and [40,50) are equal (see Supplementary Fig. 4 for the absolute numbers of vaccinated persons). The coverage for ages [0,20) is zero. The vaccination rollout data based on fully vaccinated persons[1] are shown in **b** as red dots.

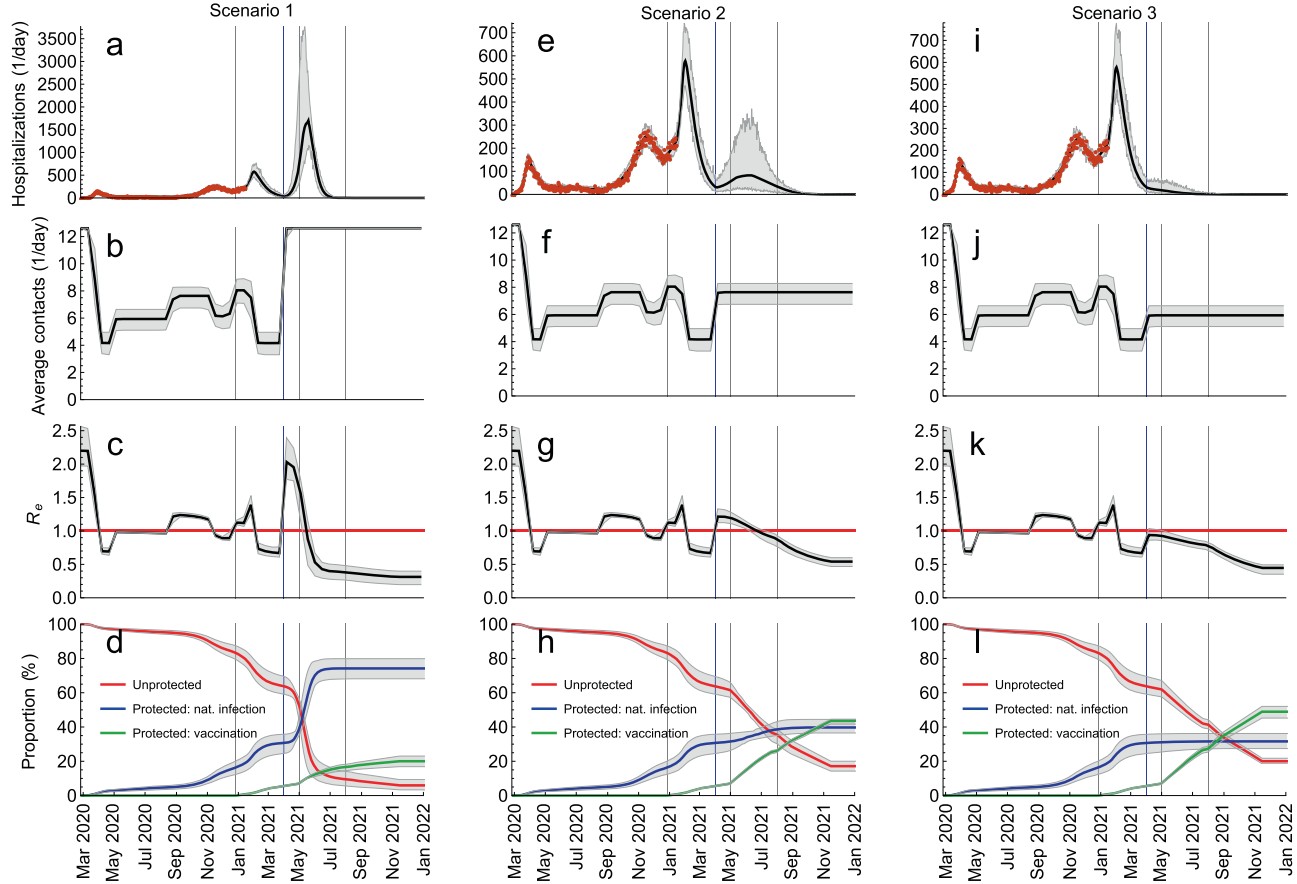

**Fig. 6 Scenarios for relaxation of control measures. a–d** Lifting all measures so that contact rates in the population return to the pre-pandemic level. **e–h** Partial lifting of measures so that contact rates increase to the level of September–October 2020. **i–l** Partial lifting of measures so that contact rates increase to the level of June–August 2020. The blue vertical lines indicate the mid-point of the transition (1 April 2020). The gray vertical lines indicate the starting dates for different vaccination phases (Table 1). The red horizontal line denotes $R_e = 1$. The hospitalization data are shown as red dots. The thick solid lines are the median trajectories estimated from the model. The gray shaded regions correspond to 95% credible intervals.

stays below 1 (Fig. 6k) but, like in Scenario 2, the measures have to continue until sufficient number of people acquire protection by vaccination to relax them completely.

In addition, we explored Scenario 4 (Fig. 7) where measures are relaxed in a step-wise manner so that contact rates first rise to the level of June–August 2020 (Step 1, Scenario 3), then to the level of September–October 2020 (Step 2, Scenario 2) and, finally, to the pre-pandemic level (Step 3, Scenario 1) (Fig. 7b). The mid-points of transitions were 1 April, 1 June and 1 October 2021 (blue vertical lines in Fig. 7) and the relaxation speed of 10 days was

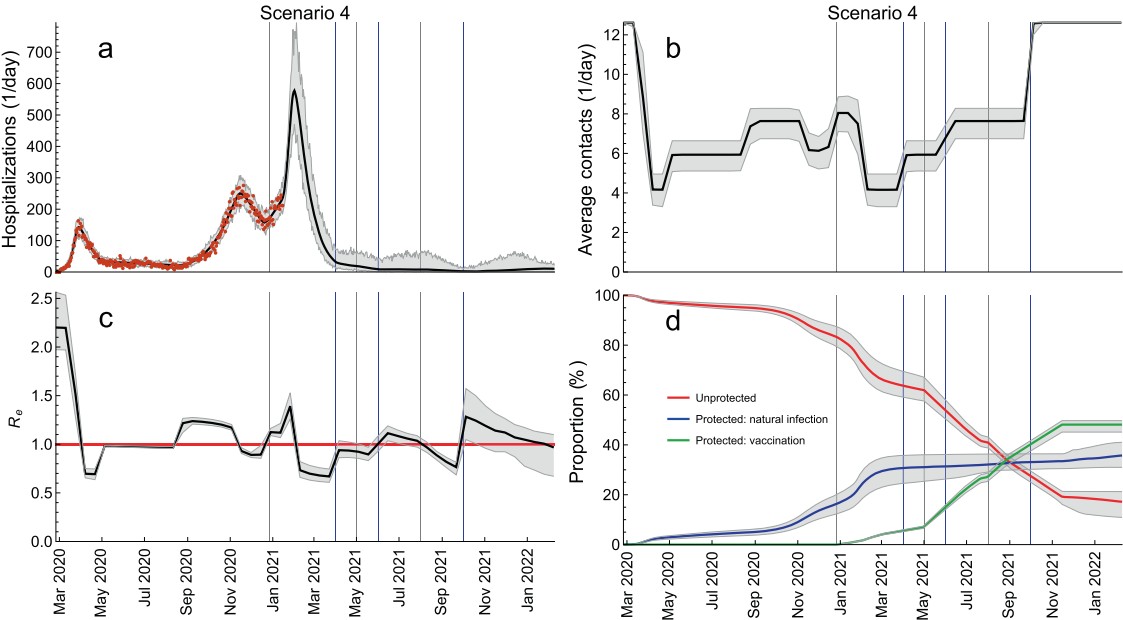

**Fig. 7 Sequential relaxation of control measures.** This scenario consists of sequential relaxation of measures so that the contact rates increase, in sequence, to the level of June–August 2020, of September–October 2020 and the pre-pandemic level. The blue vertical lines indicate the mid-points of these transitions (1 April, 1 June, 1 October). The gray vertical lines indicate the starting dates for different vaccination phases (Table 1). The red horizontal line denotes $R_e = 1$. The hospitalization data are shown as red dots. The thick solid lines are the median trajectories estimated from the model. The gray shaded regions correspond to 95% credible intervals.

used for all transitions. In this scenario, additional waves can be prevented altogether and hospitalizations stay at the level comparable to that in summer 2020 when the epidemic activity was low (Fig. 7a). The number of hospitalizations in Scenario 4 is 2.2 times larger than in Scenario 3 (3194 vs 1450 from 1 April 2021 till 1 January 2022) and 2.8 times smaller than in Scenario 2 (3194 vs 8975 in the same time period) but the situation would still seem manageable for the healthcare system because the model does not predict sharp increases in hospital admissions. Most importantly, unlike in Scenarios 2 and 3 where contact rates stay reduced after 1 April 2021, the return to pre-pandemic contact patterns in Scenario 4 is gradual and the complete lifting of measures occurs on 1 October 2021 which would have important socio-economic consequences. Interestingly, Step 2 (1 June) and Step 3 (1 October) increase $R_e$ above 1 (Fig. 7c) leading to waves of infections (Supplementary Fig. 5) but a large increase in hospitalizations is not observed because a substantial proportion of the vulnerable population has been vaccinated (Fig. 5). The full control of the pandemic ($R_e(t) < 1$ and pre-pandemic contact rates) is reached on 8 February 2022 (Fig. 7c) when 36% of the population are protected after natural infection, 48% after vaccination, and 17% stay unprotected (Fig. 7d). This is drastically different from Scenario 1, where the control was reached mainly due to protection through natural infection (60%), and the minority was protected by vaccination (10%).

We would like to stress that for demonstration purposes the timings of Steps 2 and 3 in Scenario 4 have been intentionally chosen so that the epidemic activity (i.e., the number of hospital admissions) in 2021 is similar to that in summer 2020. The premature relaxation of measures can still lead to new waves of hospitalizations. We demonstrate this in Supplementary Fig. 6 where Step 3 occurs on 1 August instead of 1 October 2021. In this case, a large outbreak is observed from August till December 2021 and the total number of hospitalizations is 3 times larger (9650 vs 3194 from 1 April 2021 till 1

January 2022) than if measures were completely lifted on 1 October 2021.

Similarly, the results presented for all scenarios are the most optimistic in terms of projected hospitalizations and get worse for a pessimistic set of vaccine efficacies. For Scenario 4 which is probably the most realistic scenario for the future relaxation of measures, the model predicts a new pandemic wave that continues until summer 2022 resulting in a 7.3-fold increase in hospitalizations (30,028 vs 4088 admissions between 1 April 2021 and 24 June 2022) for pessimistic assumptions about vaccine efficacies (Supplementary Fig. 7 and Supplementary Table 2).

We have also explored the impact on hospitalizations of behavior compensation post-vaccination by which we imply that individuals return to pre-pandemic contact rates immediately upon getting vaccinated (Supplementary Figs. 7, 8 and Supplementary Table 2). Both in the presence and in the absence of behavior compensation in Scenario 4, the number of break-through cases after vaccination is relatively small (about 5–6% of the total cumulative cases) for optimistic vaccine efficacies and is comparable to the number of infections in the unvaccinated population (about 42–46% of the total infections) for pessimistic vaccine efficacies (Supplementary Fig. 8 and Supplementary Table 2). Overall, the change in behavior of vaccinated persons would have relatively little impact on cumulative hospital admissions, i.e., an increase of 5% from 4088 to 4301 for optimistic vaccine efficacies and an increase of 4% from 30,028 to 31,344 for pessimistic vaccine efficacies (Supplementary Fig. 7 and Supplementary Table 2). Therefore, the model projections for hospital admissions depend more strongly on our assumptions regarding pessimistic and optimistic vaccine efficacies (i.e., several fold increase in hospitalizations for the range explored) and to a smaller extent on the assumptions regarding behavioral changes in the vaccinated population (i.e., few percent increase in hospitalizations for the range explored). These findings are sensitive to the infectivity of breakthrough cases in vaccinated persons (Supplementary Fig. 11 and Supplementary Table 6). In

particular, for a vaccine that is highly effective in reducing susceptibility (94% under our optimistic assumptions), the impact of infectivity of vaccinated persons on hospitalizations is small (a 15% decrease from 4088 to 3492 admissions in a hypothetical best-case scenario of zero infectivity in Scenario 4; Supplementary Table 6). For a vaccine with a low efficacy in reducing susceptibility (55% under our pessimistic assumptions), the impact on hospitalizations is much larger (a 77% decrease from 30,038 to 6410 admissions for the same scenario; Supplementary Table 6).

Finally, our results are also dependent on the assumed maximum coverage of 90% in all age groups. In reality, as vaccination coverage in older age groups will start to saturate, younger people might have lower intent to get vaccinated. The sensitivity analyses for the maximum coverage decreasing with age (90% in 80+, 85% in [50,80), 75% in [20,50) and 0% in [0,20) years old; Supplementary Figs. 8 and 9) show that the cumulative median number of hospitalizations from 1 April 2021 till 1 January 2022 for Scenarios 1, 2 and 3 would be almost equal to the situation when the maximum vaccination coverage is independent of age and very high (Figs. 5 and 6). For Scenario 4, the number of hospitalizations would be 8% higher than in Fig. 6. The reason for this is that in Scenario 1 the pandemic unfolds much faster than vaccination is rolled out. In Scenarios 2 and 3 the pandemic is partially controlled through the measures, i.e., the contact rates continue to be reduced till the end of 2021. In Scenario 4, though, the contact rates return to the pre-pandemic levels on 1 October 2021 while the coverage is not sufficiently high leading to a slight increase in hospitalizations at the end of 2021.

## Discussion

In this study, we used an age-structured model for SARS-CoV-2 transmission to generate and evaluate scenarios for relaxation of control measures during the ongoing vaccination rollout in Portugal. In agreement with the plans of the Portuguese government, the mid-point of easing of measures is April 2021. Our analyses demonstrate that vaccination alone, if rolled out according to the national vaccination schedule, is likely to be insufficient to control the Portuguese pandemic in case control measures are significantly alleviated already in April 2021. In fact, in our analyses returning to the pre-pandemic lifestyle already in spring 2021 is a worst-case scenario that would likely lead to overburdening of the healthcare system. Even for the most optimistic model assumptions, this scenario would result in a wave of hospitalizations 20% larger than the three previous waves combined together (58,226 cumulative median hospitalizations from 1 April 2021 till 1 January 2022 versus 48,273 hospitalizations from 25 February 2020 till 31 March 2021). Relaxing measures to the same extent as in autumn 2020 would lead to a smaller wave (as compared to the worst-case scenario and even to the third wave that actually occurred) that would, nonetheless, present a significant burden for the healthcare system. Our findings are qualitatively similar to those in modeling studies for China[29] and the UK[27,28]. However, a quantitative comparison is not possible because of different settings and contexts in which these studies were conducted. Additional waves could be prevented altogether if measures in spring 2021 are relaxed to the same extent as in summer 2020 or in a step-wise manner throughout 2021.

The point at which the pandemic is brought under full control ($R_e(t) < 1$ and pre-pandemic contact patterns) depends on the amount of protection in the population acquired through a combination of natural infection and vaccination. Gaining the control quickly (by mid-May 2021) occurs mainly through

protection by natural infection (60% of the population) while the minority (10%) would be protected by vaccination. As mentioned above, this worst-case scenario is, obviously, undesirable and is not very much different from letting the pandemic develop without any control measures. In the gradual relaxation scenario, achieving control takes more than one year since the start of vaccination rollout, but almost 50% of the population are protected by vaccination and a smaller proportion (35%) have experienced SARS-CoV-2 by that point. Alternative to these scenarios would be accelerating the vaccination campaign so that vaccination coverage increases faster than initially projected and confirmed by the vaccination rollout data[1]. However, it is not clear whether this option is viable for Portugal given the current shortage for COVID-19 vaccines.

In comparison with the previous studies[27–29], a strength of our analyses is that we calculate the effective reproduction number using the estimated current levels of age-specific seroprevalence and vaccination coverage in the population instead of reducing the value of $R_e$ at the beginning of the pandemic homogeneously across age groups. Another strength of our analyses is that, unlike earlier studies for China and the UK[27–29], the parameters of our model are based on formal statistical inference to match the course of the Portuguese pandemic as reflected by age-specific hospital admissions and age-specific seroprevalence data[59]. In addition, our fitting procedure allows for estimation of temporal changes in age-dependent contact patterns as a response to prior control measures during this pandemic. Therefore, instead of modeling specific relaxation policies that are not straightforward to implement in a compartmental model like ours (e.g., increased contact tracing[33]) or other policies in which governments might be interested but that do not have immediate interpretation in terms of (setting-specific) contact matrices (e.g., banned gatherings of more than 3 people and family members of COVID-19 patients have to stay at home or allowing a visitor per 25 square meters of space in a shopping area without prior appointment), we model several scenarios using the estimated contact structure after relaxation of measures in summer and autumn 2020.

In light of these past measures, our findings are easy to interpret and contain an important message for local policy-makers. School opening is thought to be the main driver of the changes observed in autumn 2020, although an increase in socializing indoors in general caused by weather alone must also have played a role. If the relaxation planned for April 2021 includes school reopening in full after Easter and resuming indoor service in restaurants and bars, then it is very likely that the average contact rate in the population will reach levels very similar to those in autumn 2020. As a consequence, this might lead to a new wave of hospitalizations as illustrated in Scenario 2. On the bright side, according to our analysis the goal of Scenario 3, in which major waves are avoided, seems well within reach, given the light control measures that were in place during summer 2020. Combining these with some additional limitations of indoor social activities and online classes for secondary school students could help to replicate the average contact rate of summer 2020, compensating for opening of elementary schools.

As any model, our model has limitations. An important one is that protection against (re-)infection after natural infection and vaccination is permanent over the time-scale of our analyses (almost two years). This frequently used assumption[27–29,45,48] leads to that in our model, theoretically, SARS-CoV-2 can be eliminated from the population. However, as we discussed recently[60] and as addressed in several conceptual modeling studies[61–63], accumulating evidence suggests that after the initial pandemic phase SARS-CoV-2 is likely to be transitioning to endemicity and continued circulation. Specifically, recent data

from individual-level studies point to that detectable levels of antibodies to SARS-CoV-2 providing immunity against reinfection can wane on the time scale of a few months to few years following exposure, as shown by our group[64] and corroborated with findings of other studies[65–67]. However, the immunity to SARS-CoV-2 depending on a combination of B- and T-cell-mediated responses elicited during primary SARS-CoV-2 infection could reduce susceptibility to and infectiousness of the following infections and offer protection against severe disease, i.e., COVID-19[68]. The estimation of the model parameters and evaluation of relaxation strategies in light of waning of sterilizing immunity lies outside the scope of our study but it should be addressed in future work when convincing data on reinfections in unvaccinated and vaccinated individuals become available.

Another limitation that deserves mention is that our results are based on early data on the efficacy in clinical trials and real-world effectiveness of the Pfizer-BioNTech vaccine[16,19–23]. We also assume that vaccine efficacy against the B.1.1.7 variant circulating in Portugal is the same as the efficacy reported from studies conducted in other locations, e.g., the recent study among working age adults in England[23], where the dominant variant in circulation was B.1.1.7. This study demonstrated that effectiveness of the Pfizer-BioNTech vaccine against symptomatic and asymptomatic infection is 86% seven days after two doses[23]. However, SARS-CoV-2 mass vaccination programmes and prolonged control measures can generate selection pressure leading to viral adaptation, antigenic divergence or vaccine escape. Viral adaptations may contribute to decreasing efficacy of existing vaccines via faster waning of (sterilizing) immunity. For example, recent experiments demonstrate that the South African variant B.1.351 shows reduced neutralizing antibody binding increasing the prospects of reinfection and hampering the efficacy of spike-based vaccines[69]. This will need consideration in vaccine development and evaluation of future vaccination programmes and relaxation scenarios in mathematical transmission models. A possible case where an antigenic escape variant caused a resurgence of COVID-19 despite high population-level seroprevalence was observed in Manaus, Brazil[31]. In Portugal, the P.1 (Brazilian) variant of concern associated with the outbreak in Manaus does not appear to be on the rise by the end of March 2021. Should this variant start to spread later during 2021, the relaxation scenarios performed for pessimistic vaccine efficacies would be more appropriate in accordance with recent experimental studies demonstrating that the P.1 variant may evade neutralizing antibody responses induced by infection and vaccination[70].

Lastly, our analyses assume a causal relation between the control measures and the reduction in circulation of SARS-CoV-2, and do not incorporate seasonal variation in transmissibility. The data on human coronaviruses (229E, HKU1, NL63, OC43) from other locations (e.g., New York or Stockholm) show a marked seasonal pattern[61,62,71] with hardly any circulation in summer. If the seasonality played a major role in transmission of SARS-CoV-2 in summer 2020 then, irrespective of the relaxation of control measures, we could expect low epidemic activity near the end of spring 2021 too. The seasonality would also imply that the effective reproduction number is higher during the winter season and lower during the summer season, and could explain a relatively low basic reproduction number (median value of 2.20) estimated by the model in March 2020, although this value lies within the range of published estimates for other countries[72].

To summarize, our study provides timely input into the discussion about the pandemic response during the vaccination rollout in Portugal. Our analyses suggest that the pressing need to restart socioeconomic activities might lead to new waves of hospitalizations in 2021 and that substantial measures prove necessary to control COVID-19 throughout 2021. More favorable scenarios that help to avoid future waves include relaxation of measures as in summer 2020 or a step-wise approach when measures are relaxed gradually until the end of 2021.

## Methods

**Overview.** The transmission model was calibrated using a combination of behavioral, surveillance and demographic data for Portugal. Parameter estimates were obtained from the model fit to (i) age-stratified COVID-19 hospitalization data ($n = 28,482$) in the period from 26 February 2020 till 15 January 2021 and (ii) cross-sectional age-stratified SARS-CoV-2 seroprevalence data ($n = 2301$) assessed from 21 May 2020 till 8 July 2020[59]. The model was further used to investigate relaxation scenarios as vaccination is rolled out in 2021.

**Data.** The hospitalization data included $n = 28,482$ COVID-19 hospitalizations longer than 24 hours by date of admission and stratified by age during the period of 325 days following the first official case in Portugal (2 March 2020). The data was padded with 5 days without hospitalizations (from 26 February till 1 March 2020) to allow for the estimation of the number of infected individuals at the start of the pandemic. The hospitalization data spanned the first wave in spring 2020, relatively low epidemic activity in summer 2020, the second wave that started in autumn 2020 till mid-December 2020 and the third wave that started in mid-December 2020 and was still ongoing on 15 January 2021. The data source for hospital data was the Central Administration of the Health System and the Shared Services of the Ministry of Health, covering all public hospitals in Portugal receiving COVID-19 patients. Since early in the pandemic, Portugal adopted a policy of hospitalizing only patients who did not gather minimum conditions for being followed at the domicile, either due to clinical or sanitary conditions. This policy has not changed during the course of the pandemic.

The SARS-CoV-2 seroprevalence data was based on the First National Serological Survey (ISNCOVID-19) in Portugal in May/July 2020[59]. This cross-sectional seroepidemiological survey was conducted on a sample of $n = 2301$ Portuguese residents, aged 1 year or older, after the first wave. The survey sample was selected using a two-stage stratified non-probability sampling design (quota sampling)[59]. SARS-CoV-2 IgM and IgG antibodies were measured in serum samples by enzyme-linked immunosorbent assay. Further details of the study are given in[59]. For the model fitting, we used the sample size, the number of positive samples and 95% confidence intervals stratified by age group reported in[59].

The demographic composition of the Portuguese residents was taken for 2019 from the Contemporary Portugal Database (Pordata)[73]. The vaccination analyses made use of the vaccination programme (Table 1), as defined by the Directorate-General of Health prior to the start of the vaccination campaign[54]. The programme defines vaccine uptake prioritization by age and morbidities and runs in three phases from 27 December 2020 till 31 December 2021. The age distribution of morbidities in the Portuguese population was extracted from the Shared Services of the Ministry of Health on the basis of ICPC-2 (International Classification of Primary Care) codes (Supplementary Table 3). The vaccination rollout data for Portugal was taken from[1].

The baseline (pre-pandemic) contact matrices for transmission-relevant contacts for Portugal were taken from the recent study by Mistry et al.[74]. The contact matrix for Portugal after the introduction of measures to control the first wave of hospitalizations (April 2020) was inferred using the contact matrix for the Netherlands based on a cross-sectional survey carried out in April 2020 (PIENTER Corona study)[75].

**Transmission model.** We extended an age-stratified SARS-CoV-2 transmission model from[43] to include vaccination (Fig. 8). The model has susceptible-exposed-infectious-recovered structure, whereby susceptible persons ($S$) may become latently infected ($E$) before progressing to become infectious ($I$). The latently infected persons are infected with SARS-CoV-2 but not yet infectious. Persons enter the $I$-compartment when they become infectious independently of whether they have symptoms or not. Therefore, this compartment contains both symptomatic and asymptomatic individuals. The stratification in these two categories is not done because the parameters of the model would not be identified from the model fit to the data streams we used. Note that in our previous model[43], we split the infectious period into several stages, thereby obtaining a more realistic distribution of the infectious period (i.e., Erlang/Gamma distribution instead of an exponential distribution used here). For computational reasons (much longer time series and more complex projections) we do not implement an Erlang-distributed infectious period this time. Infectious persons either get hospitalized ($H$) or recover without hospitalization ($R$). Disease-related mortality and discharge from the hospital are not explicitly modeled. Therefore, the $H$-compartment contains the cumulative number of persons who experience severe symptoms and recover (or die) after admission to the hospital. Similarly, the $R$-compartment contains the cumulative number of persons who recover after having mild or no symptoms. The force of infection is given by a weighted sum of the fraction of the infectious population in different age groups (red dashed boxes in Fig. 8). We consider a

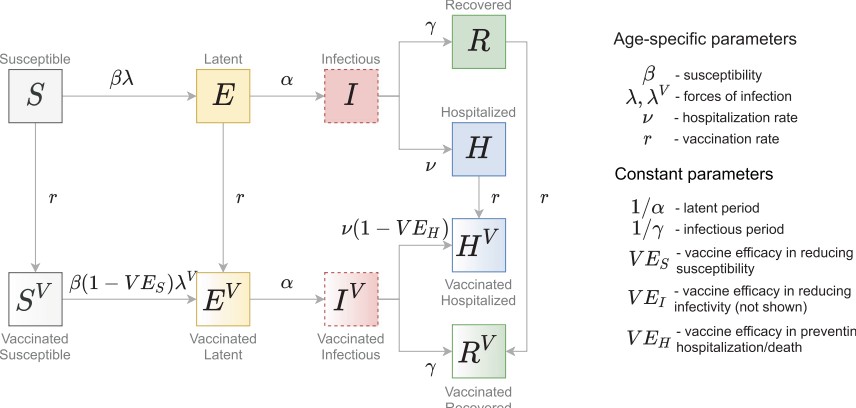

**Fig. 8 Schematic of the transmission model.** Gray arrows show epidemiological transitions. Red dashed boxes indicate compartments contributing to the forces of infection. The model is age-structured and involves an extended SEIR-type framework. Vaccinated persons may experience behavior compensation post-vaccination modeled as a return to pre-pandemic contact rates among vaccinated persons as compared to unvaccinated persons who may continue to have reduced contact rates due to control measures. The vaccine has three effects: (i) reduction in susceptibility of vaccinated relative to unvaccinated ($VE_S$); (ii) reduction in infectivity of vaccinated relative to unvaccinated ($VE_I$, see Eqs. (3) and (4)); (iii) reduction in hospitalization rate of vaccinated relative to unvaccinated ($VE_H$).

stable population and thus do not include natural birth and death processes. The contact rates, forces of infection, susceptibilities and hospitalization rates are age-specific.

In line with the current guidelines, we assume that vaccine can be delivered to all people independently from their disease history with the exception of those who might be currently infectious (I-compartment). Not vaccinating infectious compartment implies that vaccine is not given to asymptomatic persons but these represent a small fraction of the population at any given time as the absolute majority of the population is either in susceptible (S-compartment) or in recovered states (H and R-compartments). We also vaccinate the H-compartment as this compartment comprises everyone who has ever been admitted to hospital. Whilst this assumption means that the currently hospitalized persons are vaccinated too, their number is very small compared to the total number of people in the H-compartment. The vaccine has three mechanisms of action: (i) reducing susceptibility ($VE_S$); (ii) reducing infectivity ($VE_I$); (iii) reducing hospitalization rate ($VE_H$). The vaccine has no effect in persons who recovered from natural infection (R and H compartments). We assume that protection after vaccination is achieved immediately and is equivalent to two vaccine doses, and that the duration of protection after both natural infection and vaccination is about two years (time horizon of our analyses). Finally, we allow for behavior compensation post-vaccination modeled as a return to pre-pandemic contact rates among vaccinated persons as compared to unvaccinated persons who may continue to have reduced contact rates due to control measures. This is reflected in generally different forces of infection for unvaccinated and vaccinated persons. The full description of the model parameters is given in Supplementary Tables 1 and 5.

**Model equations.** The model was implemented in Mathematica 10.0.2.0 using a system of ordinary differential equations for the number of persons in different compartments shown in Fig. 1. The transmission model was stratified into $n = 10$ age groups: [0, 5), [5, 10), [10, 20), [20, 30), [30, 40), [40, 50), [50, 60), [60, 70), [70, 80), 80+.

The equations for the numbers of unvaccinated persons in age group $k$, $k = 1$, ..., $n$, who are susceptible ($S_k$), exposed ($E_k$), infectious ($I_k$), recovered ($R_k$) and hospitalized ($H_k$) read as follows

$$\begin{aligned}
\frac{dS_k(t)}{dt} &= -\beta_k \lambda_k(t) S_k(t) - \frac{r_k S_k(t)}{S_k(t) + E_k(t) + R_k(t) + H_k(t)}, \\
\frac{dE_k(t)}{dt} &= \beta_k \lambda_k(t) S_k(t) - \alpha E_k(t) - \frac{r_k E_k(t)}{S_k(t) + E_k(t) + R_k(t) + H_k(t)}, \\
\frac{dI_k(t)}{dt} &= \alpha E_k(t) - (\gamma + \nu_k) I_k(t), \\
\frac{dR_k(t)}{dt} &= \gamma I_k(t) - \frac{r_k R_k(t)}{S_k(t) + E_k(t) + R_k(t) + H_k(t)}, \\
\frac{dH_k(t)}{dt} &= \nu_k I_k(t) - \frac{r_k H_k(t)}{S_k(t) + E_k(t) + R_k(t) + H_k(t)}.
\end{aligned} \tag{1}$$

The equations for the numbers of vaccinated persons in age group $k$ who are vaccinated susceptible ($S_k^V$), exposed ($E_k^V$), infectious ($I_k^V$), recovered ($R_k^V$) and

hospitalized ($H_k^V$) are given by

$$\begin{aligned}
\frac{dS_k^V(t)}{dt} &= -\beta_k(1 - VE_S)\lambda_k^V(t)S_k^V(t) + \frac{r_k S_k(t)}{S_k(t) + E_k(t) + R_k(t) + H_k(t)}, \\
\frac{dE_k^V(t)}{dt} &= \beta_k(1 - VE_S)\lambda_k^V(t)S_k^V(t) - \alpha E_k^V(t) + \frac{r_k E_k(t)}{S_k(t) + E_k(t) + R_k(t) + H_k(t)}, \\
\frac{dI_k^V(t)}{dt} &= \alpha E_k^V(t) - (\gamma + \nu_k(1 - VE_H))I_k^V(t), \\
\frac{dR_k^V(t)}{dt} &= \gamma I_k^V(t) + \frac{r_k R_k(t)}{S_k(t) + E_k(t) + R_k(t) + H_k(t)}, \\
\frac{dH_k^V(t)}{dt} &= \nu_k(1 - VE_H)I_k^V(t) + \frac{r_k H_k(t)}{S_k(t) + E_k(t) + R_k(t) + H_k(t)}.
\end{aligned} \tag{2}$$

Persons get vaccinated in S, E, R and H states. The vaccination rates $r_k$ are age-specific. We denote the contact rate of an unvaccinated person in age group $k$ with persons in age group $l$, $c_{kl}(t)$, and the contact rate of a vaccinated person in age group $k$ with persons in age group $l$, $c_{kl}^V(t)$. The forces of infection for unvaccinated and vaccinated persons are given by

$$\lambda_k(t) = \epsilon \sum_{l=1}^{n} c_{kl}(t) \frac{I_l(t) + (1 - VE_I)I_l^V(t)}{N_l}, \tag{3}$$

$$\lambda_k^V(t) = \epsilon \sum_{l=1}^{n} c_{kl}^V(t) \frac{I_l(t) + (1 - VE_I)I_l^V(t)}{N_l}, \tag{4}$$

where $N_k$ is the number of individuals in age group $k$, $N_k = S_k(t) + E_k(t) + I_k(t) + H_k(t) + R_k(t) + S_k^V(t) + E_k^V(t) + I_k^V(t) + R_k^V(t) + H_k^V(t)$. Note that Eqs. (3) and (4) imply that the entire population participates in the contact process including persons in the H-compartment but that H-persons are not infectious. This is based on the fact that the vast majority of people in the H-compartment are recovered after hospitalization, and a very small proportion is currently hospitalized. We assume that currently hospitalized persons continue to have contacts with the personnel and visitors but they cannot infect them because of the use of individual protective measures.

The initial condition for the model was $E_k(t = 0) = I_k(t = 0) = \frac{1}{2}\theta N_k$ and $S_k(t = 0) = (1 - \theta)N_k$, where $t = 0$ is 26 February 2020. The parameter $\theta$ denotes the initial fraction of the population that was infected (split equally between infectious and exposed). This parameter accounts for importation of new cases at the start of the pandemic and was estimated jointly with other parameters. Importation of cases was not implemented at later stages of the pandemic due to a large pool of infectious individuals within the country.

The rapid spread of B.1.1.7 variant, that is estimated to be about 50% more transmissible based on the data from England[5–7], fueled the third wave of hospitalizations in Portugal. The increasing dominance of this variant was modeled empirically as a gradual increase in the probably of transmission per contact by 50% as follows $\epsilon[1 + 0.5/(1 + e^{-K_0(t - t_{\text{data}})})]$, where $\epsilon$ and $K_0$ were estimated based on the data until 15 January 2021 (Supplementary Fig. 2) and $t_{\text{data}}$ is the last date in the hospital admission data (15 January 2021).

**Observation model and parameter estimation.** To generate a set of plausible parameters and initial conditions for our projections, we fitted the model to

hospitalization data and serological testing data, using a similar approach as before[43,76]. We incorporated the transmission model, Eq. (1), in a Bayesian statistical model with likelihood function constructed as follows. Let $h_{k,m}$ denote the observed number of hospitalizations in age group $k$ and day $t_m$. The expected number of hospitalizations during day $t_m$ is approximately equal to $\overline{h}_{k,m} := \nu_k \cdot I_k(t_m)$. To account for reporting errors and heterogeneity in the hospitalization rate within age groups, we assume that $\overline{h}_{k,m}$ has a negative-binomial distribution with mean $\overline{h}_{k,m}$ and variance $\overline{h}_{k,m} \cdot (1 + \overline{h}_{k,m}/\phi)$. The parameter $\phi$ determines the overdispersion of the reporting of hospitalizations. The hospitalization data were stratified into the ten age groups [0, 5), [5, 10), [10, 20), [20, 30), [30, 40), [40, 50), [50, 60), [60, 70), [70, 80), 80+.

The seroprevalence data were stratified into the five age groups [1, 10), [10, 20), [20, 40), [40, 60) and 60+[59]. Hence, for the hospitalization data and the transmission model, a finer age stratification is used than for the seroprevalence data. We assume that individuals in seroprevalence age group $G_i^s$ were sampled from hospitalization age class $G_k^h$ with probability $p_{ik}$ proportional to the relative population size of $G_k^h$ compared to $G_i^s$, i.e.,

$$p_{ik} = N_k/N_i^s, \quad \text{where} \quad N_i^s = \sum_{\ell:G_\ell^h \subseteq G_i^s} N_\ell. \tag{5}$$

As before[43], we assume that the seroprevalence data represents a random sample from each age group. Hence, the number of positive samples $\ell_i$ has a binomial distribution with population size $L_i$, equal to the total number of samples for age class $i$, and success probability $q_i$. The success probability is defined in terms of the fraction of susceptible individuals $S_k(T)$ at sampling time $T$ and the probabilities $p_{ik}$:

$$q_i = \sum_{k:G_k^h \subseteq G_i^s} (1 - S_k(T)/N_k)p_{ik}. \tag{6}$$

To account for the fact that no children below the age of 1 year were included in the serology samples, we reduced the population size $N_1$ with the size of the age group [0, 1) (86,579 persons) in Eqs. (6) and (5).

The prior distribution of the model is specified in Supplementary Table 4. Informative priors were used for the latent ($1/\alpha$) and infectious ($1/\gamma$) periods. Our choice of the range for the prior for the latent period (time between infection and becoming infectious) was based on the estimates of 4–6 days for the incubation period (time between infection and developing symptoms)[72,77]. Since the current evidence suggests that SARS-CoV-2 transmission is possible before the development of symptoms, the latent period was chosen to be shorter than the incubation period, i.e., we used a narrower range of 2–5 days for 99% of the prior density of the latent period. The average generation interval in the model is $1/\alpha + 1/\gamma$. The priors on $\alpha$ and $\gamma$ were chosen to match observed generation intervals (more precisely the serial interval[78]) of on average 7 to 8 days. Also note that the effective infectious period has likely decreased because (self-)isolation upon the development of respiratory symptoms has been recommended and, in certain situations, enforced (e.g., at schools, hospitals) during the course of the pandemic[78]. Contact tracing and testing of asymptomatic persons also decreases the time of infectiousness. The mean a priori generation interval was 7.3 days (99% CrI 5.5–9.3). The individual serial intervals and duration of the infection have a much wider distribution.

The model was fitted with Stan[79] in R 3.6.0 and R Studio 1.3.1056 using cmdstanr package. We used 4 parallel chains, each of length 1000, with a warm-up period of 500, resulting in 2000 samples from the posterior distribution. Convergence was assessed with the Gelman-Rubin $\hat{R}$-statistic, which was close to 1 for all parameters.

The estimated model parameters are shown in Supplementary Figs. 1 and 2. As in our previous work[43] (Supplementary Fig. 5 therein), some of the parameters such as e.g., the infectious period, the initial fraction of infected individuals, the probability of transmission per contact and the hospitalization rate are strongly positively and negatively correlated. However, the outcomes of the model such as the cumulative number of hospitalizations during the study period are not sensitive to the key epidemiological parameters among which the infectious period, the latent period and the probability of transmission per contact. See scatter plots in Supplementary Fig. 12 made for Scenario 4 and a pessimistic set of vaccine efficacies where Pearson correlation coefficients between the three parameters and cumulative hospitalizations from 25 February 2020 till 24 June 2022 are in the range of 0.09 to 0.14.

**Time-varying contact patterns.** The contact patterns in the population varied with time due to introduction/reinforcement or relaxation of control measures as follows: 0) introduction of measures to control the first pandemic wave (first lockdown, March 2020); 1) relaxation of measures after the first wave was curbed (May 2020); 2) further relaxation of measures that included school opening (September 2020); 3) reinforcement of measures to control the second wave (second lockdown, November 2020); 4) relaxation of measures around Christmas 2020; 5) reinforcement of measures to control the third wave (third lockdown, January 2021).

We denote $c_{kl}(t)$ the contact rate for a person in age group $k$ ($k = 1, ..., n$) with persons in age group $l$ ($l = 1, ..., n$) at time $t$. The contact rate denotes the number of transmission-relevant contacts per day such as touching or having a conversation with someone[74,75]. Our fitting procedure allows to estimate $c_{kl}(t)$ by

assuming that changes due to control measures described in 0)-5) occur as a series of smooth transitions.

To describe the transition 0) from the baseline (pre-pandemic) contact rate $b_{kl}$ to the contact rate after the first lockdown $a_{kl}$ we write down $c_{kl}(t)$ as a linear combination of contact rates $b_{kl}$ and $a_{kl}$ with coefficients constructed using a logistic function $f_0(t) = 1/(1 + e^{-K_0(t-t_0)})$ as follows

$$c_{kl}(t) = [1 - f_0(t)]b_{kl} + f_0(t)\zeta a_{kl}. \tag{7}$$

The parameter $K_0$ of the logistic function describes the speed with which the first lockdown is enforced. The parameter $t_0$ describes the mid-time of the introduction of the first lockdown. Note in Eq. (7) we introduced the factor $\zeta \in [0, 1]$ to reflect that not all reported contacts after the first lockdown might be relevant for transmission, for example, due to mask-wearing or physical distancing when a contact took place. Therefore, the baseline (pre-pandemic) contact rates are described by the matrix $b_{kl}$, and the contact rates after the first lockdown are described by the matrix $\zeta a_{kl}$.

The pre-pandemic matrix $b_{kl}$ for Portugal was taken from[74] (Fig. 9a). The matrix after the first lockdown $a_{kl}$ was inferred using the contact matrix for the Netherlands based on a cross-sectional survey carried out in April 2020 (PIENTER Corona study)[75]. Since measures enforced during the first lockdown in the two countries were similar (e.g., all schools were closed, all non-essential work was done from home etc.) we reduced the age-specific contact rates for Portugal after the lockdown by the same percentage as it was observed in the Netherlands (Fig. 9b). The resulting number of daily contacts for a person in given age group at baseline and after the lockdown in April 2020 is shown in Fig. 9c. Like for the Netherlands[75], we observe larger reductions in contacts for children (due to school closure) and smaller reductions for elderly because most of their contacts were essential (e.g., with healthcare personnel or caretakers) and thus were not affected by the lockdown. The parameter $\zeta$ that multiplies the inferred matrix $a_{kl}$ can account for discrepancies between the real and inferred matrix.

To describe the contact rates after transitions 1)-4) have taken place, we assume that these can be written as a linear combination $u_i b_{kl} + (1 - u_i)\zeta a_{kl}$, $i = 1, ..., 4$, where $u_i$ is the proportion of time a person behaves as before the pandemic and $(1 - u_i)$ is, respectively, the proportion of time a person behaves as during the first lockdown. This contact structure can, therefore, interpolate between the first (most strict) lockdown and no measures in place at all. Since the third lockdown was similar to the first lockdown, the transition 5) was modeled as a return to the lockdown contact matrix $\zeta b_{kl}$. As before, the transitions between the contact rates during periods 1)-5) are modeled using logistic functions $f_i(t) = 1/(1 + e^{-K_i(t-t_i)})$, where $i = 1, ..., 5$. The general contact rate can therefore be written as

$$\begin{aligned}c_{kl}(t) =& [1 - f_0(t)]b_{kl} + f_0(t)\zeta a_{kl}[1 - f_1(t)] + f_1(t)[u_1 b_{kl} + (1 - u_1)\zeta a_{kl}][1 - f_2(t)] \\ & + f_2(t)[u_2 b_{kl} + (1 - u_2)\zeta a_{kl}][1 - f_3(t)] + f_3(t)[u_3 b_{kl} + (1 - u_3)\zeta a_{kl}][1 - f_4(t)] \\ & + f_4(t)[u_4 b_{kl} + (1 - u_4)\zeta a_{kl}][1 - f_5(t)] + f_5(t)\zeta a_{kl}.\end{aligned} \tag{8}$$

All the parameters that describe $c_{kl}(t)$, except for the last transition 5) for which hospitalization data are not available, are estimated (Supplementary Table 5). The estimates for these 15 parameters $\zeta$, $u_i$ ($i = 1, ..., 4$), $t_i$ ($i = 0, ..., 4$) and $K_i$ ($i = 0, ..., 4$) are shown in Supplementary Fig. 2. The estimated logistic functions are plotted in Fig. 9d.

In the main analyses (Figs. 6 and 7), the contact rates for vaccinated persons were equal to those unvaccinated, $c_{kl}^V(t) = c_{kl}(t)$. In the sensitivity analyses (Supplementary Figs. 7, 8 and Supplementary Table 2), they were set to pre-pandemic contacts as follows, $c_{kl}^V(t) = b_{kl}$. The contact rate presented in Figs. 3, 6 and 7 was the average contact rate in the population calculated as follows $\langle c(t) \rangle = \sum_{k=1}^n \sum_{l=1}^n c_{kl}(t)N_k/\sum_{k=1}^n N_k$. Note that this expression makes use of the fact that in the main analyses $c_{kl}^V(t) = c_{kl}(t)$.

The relaxation scenarios during the vaccination rollout are modeled as a transition from the contact rate described by Eq. (8) to the contact rate $b_{kl}$ (Scenario 1); $u_2 b_{kl} + (1 - u_2)\zeta a_{kl}$ (Scenario 2); $u_1 b_{kl} + (1 - u_1)\zeta a_{kl}$ (Scenario 3); $u_1 b_{kl} + (1 - u_1)\zeta a_{kl}$ (Scenario 4, Step 1); $u_2 b_{kl} + (1 - u_2)\zeta a_{kl}$ (Scenario 4, Step 2); $b_{kl}$ (Scenario 4, Step 3). The parameters of the logistic functions describing these transitions are specified in Supplementary Table 5.

**Time-varying effective reproduction number.** The basic reproduction number, $R_0$, is the average number of secondary infections caused by a single infectious individual at the beginning of the epidemic in a disease-free, totally susceptible population. If $R_0 > 1$ the disease will spread exponentially. If $R_0 < 1$ the number of infectious persons declines exponentially and the disease is not able to spread. In general, $R_0$ depends on the type of virus but also on the contact patterns in the population.

When the disease has already spread and we have no longer a fully susceptible population but some part of the population is immune due to natural infection or vaccination, the generalization of $R_0$ is given by the effective reproduction number, $R_e(t)$. $R_e(t)$ depends on the type of virus, the level of population immunity and the contact patterns in the population. The full control of the disease is achieved when $R_e(t) < 1$ and the contact rates in the population are at their pre-pandemic levels, i.e., not anymore affected by control measures. A partial control is achieved when

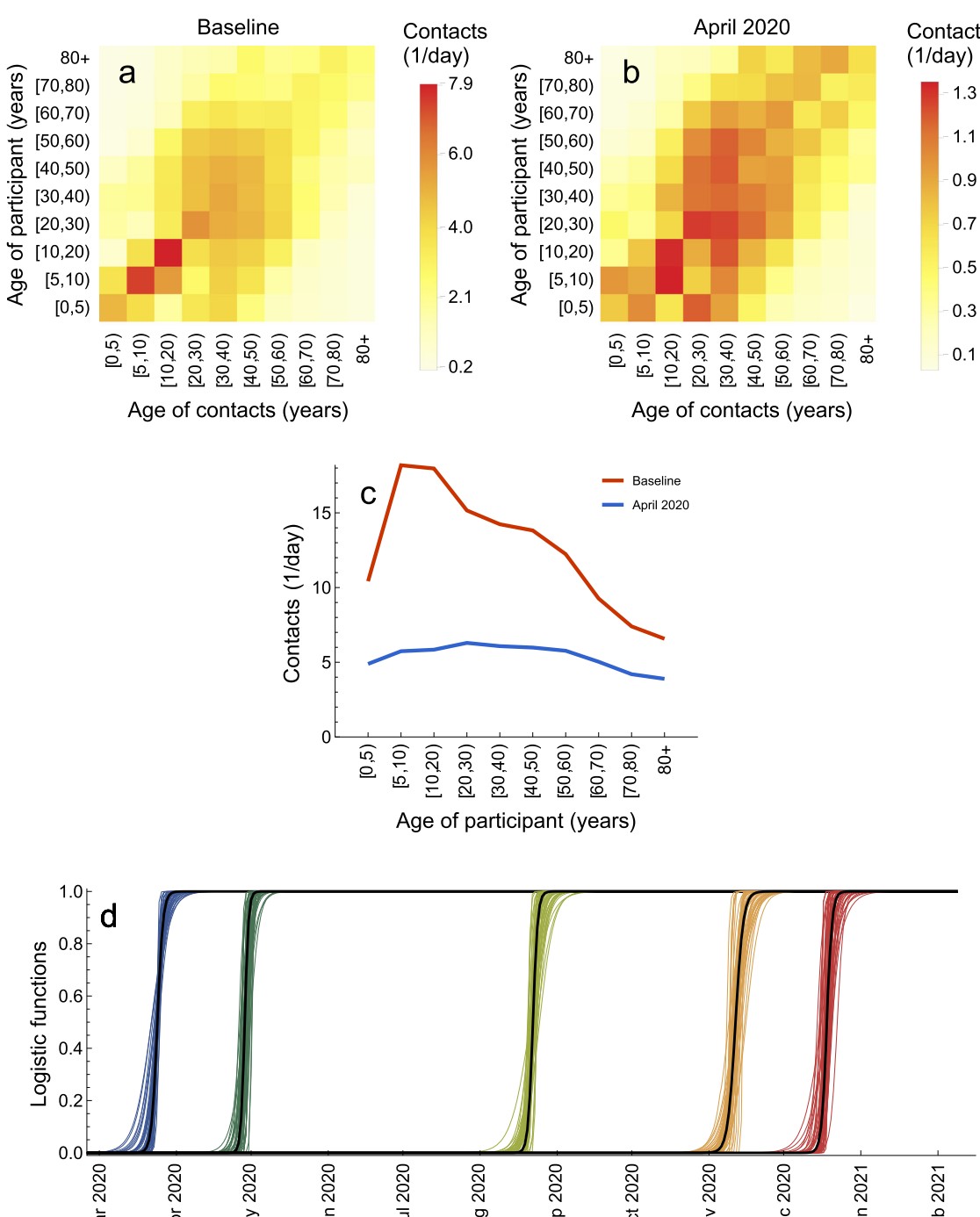

**Fig. 9 Contact matrices. a** Baseline (pre-pandemic) contact matrix. **b** Contact matrix after the introduction of measures in April 2020. **c** Average number of contacts for a person in a given age group. **d** Logistic functions describing transitions between contact matrices. Shown are $f_0$ (blue), $f_1$ (dark green), $f_2$ (light green), $f_3$ (orange), and $f_4$ (red) based on 50 samples from the posterior distribution.

$R_e(t) < 1$ but the contact rates have not been restored to their pre-pandemic levels yet as is currently the case for SARS-CoV-2 in Portugal.

In a deterministic compartmental model such as the one employed here, the calculation of $R_0$ and $R_e(t)$ can be performed using the next-generation matrix (NGM) method[80]. The starting point of the method is to calculate the Jacobian $\mathbf{J}$ of the equations for the latent ($E_k$, $E_k^V$) and infectious ($I_k$, $I_k^V$) age classes $k$, $k = 1, \ldots, n$, isolated from the full model given by Eqs. (1) and (2). The Jacobian $\mathbf{J}$ is then evaluated at the disease-free equilibrium of interest.

For $R_0$ calculation, the disease-free equilibrium is

$$S_k^* = N_k, \qquad S_k^{V*} = E_k^* = E_k^{V*} = I_k^* = I_k^{V*} = 0, \qquad k = 1, \ldots, n. \qquad (9)$$

For $R_e(t)$ calculation with or without vaccination, the disease-free equilibrium is

$$S_k^* = S_k(t), \qquad S_k^{V*} = S_k^V(t), \qquad E_k^* = E_k^{V*} = I_k^* = I_k^{V*} = 0, \qquad r_k = 0, \qquad k = 1, \ldots, n, \qquad (10)$$

where the time-dependent variables $S_k(t)$ and $S_k^V(t)$ are obtained from the solutions of the full model given by Eqs. (1) and (2).

Following[80], the Jacobian $\mathbf{J}$ may be recast as follows

$$\mathbf{J} = \mathbf{T} + \mathbf{\Sigma}, \qquad (11)$$

where the transmissions matrix $\mathbf{T}$ contains the terms associated with the

production of new infections, and the transitions matrix $\mathbf{\Sigma}$ contains the terms associated with all other state changes. After performing this operation, we construct a new matrix $\mathbf{K_L}$, called the large domain NGM[80], given by

$$\mathbf{K_L} = -\mathbf{T\Sigma}^{-1}. \tag{12}$$

The basic reproduction number $R_0$ at time $t = 0$ and the effective reproduction number $R_e(t)$ at any time $t$ are given by the spectral radius of $\mathbf{K_L}$ which is the largest eigenvalue of $\mathbf{K_L}$. For the purpose of computing the spectral radius, $\mathbf{K_L}$ can be further reduced as detailed in[80]. The explicit expressions for matrices $\mathbf{J}$, $\mathbf{T}$, $\mathbf{\Sigma}$ and $\mathbf{K_L}$ are given in the Mathematica notebooks available in the GitHub repository, https://github.com/lynxgav/COVID19-vaccination[57].

**Population immunity**. The unprotected population was computed as the number of individuals in the fully susceptible compartment $S$ (Fig. 8). The population protected by natural infection was computed as all individuals arriving into the infectious compartment $I$, independently of whether these individuals will or will not be vaccinated later on. Recall, that in the model vaccine has no effect in individuals who are recovered from natural infection and, therefore, the population protected by vaccination grows slower than vaccination coverage. The population protected by vaccination was computed as all individuals arriving into the compartments $S^V$ and $E^V$ due to vaccination.

**Vaccine efficacies**. Vaccine efficacies in reducing susceptibility ($VE_S$), infectivity ($VE_I$) and hospitalization rate ($VE_H$) were set using initial data from clinical trials and real-word studies for the Pfizer-BioNTech vaccine[16,19–23]. Important to note, that the efficacies reported in all these studies are not conditioned on infection while they are in the models like ours. For a more complete discussion on this topic, we refer the reader to the pedagogical work by Lipsitch and Kahn[24] and the report for England by the Scientific Advisory Group for Emergencies[27].

The vaccine efficacy in reducing susceptibility ($VE_S$) was set based on vaccine efficacies and effectiveness against infection ($VE_\mathrm{infection}$) reported in clinical trials and real-word studies, i.e.,

$$VE_\mathrm{infection} \equiv VE_S. \tag{13}$$

The vaccine efficacy in reducing infectivity ($VE_I$) was assumed to be the same as vaccine efficacy in reducing disease conditioned on infection ($VE_\mathrm{disease|infection}$), i.e., $VE_\mathrm{disease|infection} \equiv VE_I$. $VE_\mathrm{disease|infection}$ was calculated using the efficacy against disease ($VE_\mathrm{disease}$) reported in clinical trials as follows

$$VE_\mathrm{disease} = VE_\mathrm{infection} + (1 - VE_\mathrm{infection})VE_\mathrm{disease|infection}. \tag{14}$$

The vaccine efficacy in reducing hospitalization rate ($VE_H$) is equal to vaccine efficacy against severe disease conditioned on disease ($VE_\mathrm{severedisease|disease}$), i.e., $VE_\mathrm{severedisease|disease} \equiv VE_H$. $VE_\mathrm{severedisease|disease}$ was calculated using the vaccine efficacy against severe disease ($VE_\mathrm{severe\ disease}$) reported in trials as follows

$$\begin{aligned} VE_\mathrm{severedisease} &= VE_\mathrm{infection} + (1 - VE_\mathrm{infection})VE_\mathrm{disease|infection} \\ &+ (1 - VE_\mathrm{infection})(1 - VE_\mathrm{disease|infection})VE_\mathrm{severedisease|disease}. \end{aligned} \tag{15}$$

We used an optimistic and a pessimistic set of vaccine efficacies for $VE_S$, $VE_I$ and $VE_H$ (Supplementary Table 1) based on the range of values for $VE_\mathrm{infection}$, $VE_\mathrm{disease}$, and $VE_\mathrm{severe\ disease}$ reported in the literature[16,19–23]. For the optimistic set explored in the main analyses (Figs. 6 and 7), we used $VE_\mathrm{infection} = 94\%$, $VE_\mathrm{disease} = 94\%$, and $VE_\mathrm{severe\ disease} = 98\%$ (corresponding to $VE_S = 94\%$, $VE_I = 0\%$, and $VE_H = 67\%$)[16,19,20,22,23,27]. For the pessimistic set explored in sensitivity analyses (Supplementary Fig. 7), we used $VE_\mathrm{infection} = 55\%$, $VE_\mathrm{disease} = 55\%$, and $VE_\mathrm{severe\ disease} = 55\%$ (corresponding to $VE_S = 55\%$, $VE_I = 0\%$, and $VE_H = 0\%$)[20,21,27]. Other efficacies reported in the literature for the Pfizer-BioNTech vaccine and other existing vaccines fall in between the optimistic and pessimistic values we used. This broad range of values is also relevant in case the market share of different vaccine brands in Portugal gets changed throughout 2021.

Note that both the optimistic and pessimistic sets imply that the infectivity of breakthrough cases in vaccinated persons is the same as infectivity of cases in unvaccinated persons ($VE_I = 0\%$). Therefore, we included sensitivity analyses for this parameter (Supplementary Fig. 11 and Supplementary Table 6) by taking two additional values (Supplementary Table 1): $VE_I = 50\%$ that corresponds to 50% infectivity of vaccinated persons relative to infectivity of unvaccinated persons and $VE_I = 100\%$ that is a best-case scenario implying that breakthrough cases are not infectious at all.

**Summary of sensitivity analyses**. The sensitivity analyses have been conducted with respect to (i) timings of different relaxation steps where Step 3 occurs on 1 August instead of 1 October 2021 (Supplementary Fig. 6; Scenario 4); (ii) pessimistic set of vaccine efficacies (Supplementary Figs. 7, 8 and Supplementary Table 2; Scenario 4); (iii) sensitivity to infectivity of breakthrough cases in vaccinated persons (Supplementary Fig. 11 and Supplementary Table 6); (iv) behavior compensation post-vaccination modeled as a return to pre-pandemic contact rates among vaccinated persons as compared to unvaccinated persons who may continue to have reduced contact rates due to control measures (Supplementary Figs. 7, 8 and Supplementary Table 2; Scenario 4); (v) the maximum vaccination coverage in different age groups decreasing with age, i.e., the coverage for ages

[0,20), [20,50), [50,80), and 80+ was 0%, 75%, 85%, and 90%, respectively (Supplementary Figs. 9 and 10; Scenarios 1, 2, 3 and 4). In all cases, the comparison was done based on the cumulative median number of hospitalizations after the start of the relaxation of measures until the end of the study period.

**Reporting summary**. Further information on research design is available in the Nature Research Reporting Summary linked to this article.

## Data availability

All datasets analyzed and generated in this study are publicly available at https://github.com/lynxgav/COVID19-vaccination[57].

## Code availability

The codes reproducing the results of this study are publicly available at https://github.com/lynxgav/COVID19-vaccination[57].

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

## Acknowledgements

G.R., J.V., A.N., M.C.G. were supported by Fundação para a Ciência e a Tecnologia (FCT) project reference 131_596787873, awarded to G.R. M.V. was supported by the European Union H2020 ERA project (No. 667824 - EXCELLtoINNOV). The contribution of C.H.v.D. was under the auspices of the US Department of Energy (contract number 89233218CNA000001) and supported by the National Institutes of Health (grant number R01-OD011095). MK acknowledges support from the Netherlands Organization for Health Research and Development (ZonMw) Grant no. 10430022010001.

## Author contributions

G.R. conceived and supervised the study. G.R. and J.V. developed the transmission model. C.H.v.D. developed the observation model. J.V. conducted preliminary model analyses. G.R. conducted all final analyses, prepared figures and wrote the manuscript. A.N., M.C.G., M.v.B., M.E.K., and M.V. provided data, validated the model and analyses. All authors contributed to interpretation of the results, editing the manuscript and gave final approval for publication.

## Competing interests

The authors declare no competing interests.
