## [Peer Review File · Nature Communications]

Reviewers' Comments:

Reviewer #2:

Remarks to the Author:

The paper studies effects of mass vaccination against SARS-CoV-2 (using Portugal as a case study), and considers several possible relaxation scenarios, feasible under the circumstances. The study is based on an age-structured transmission model calibrated to age-specific seroprevalence and hospital admissions data, as well as the projected vaccination coverage for Portugal. The impact of each considered scenario is quantitatively evaluated with respect to projected hospital admissions and the resultant effective reproduction number. Importantly, the paper distinguishes the achievable levels of the population immunization arising due to either natural infection or vaccination. The comparative analysis allows the Authors to quantify projected timing of controlling COVID-19 in Portugal.

The model used in this study has been partially validated previously, submission's ref. [43]. This work extends the model in several ways, most notably in considering effects of vaccination, by augmenting the model by several compartmental components accounting for vaccination (S^V , E^V , I^V , R^V and H^V - by the way, superscript V is missing in H^V in Fig. 8). The extended model retains the concept of cross-age contact interactions between the age groups, which is a strong point of the approach, making it much more refined than the traditional compartmental models, and increasing the resolution, while maintaining the power of analytical approaches. As such, it offers a clear way to combine the strengths of both compartmental models and higher-resolution agent-based models.

Nevertheless, as it is essentially a new model (not least, in terms of vaccination parameters), new sensitivity analysis needs to be performed and reported on. I was not able to find the outcomes of sensitivity analysis in the previous study (i.e., [43]), and some comments on the sensitivity of the model to changes in key epidemiological parameters, such as latent and infectious periods, would be beneficial. While the ranges of these parameters are provided and are within acceptable bounds, how robust is the model and its outcomes is unclear.

Even more importantly, the sensitivity analysis needs to be performed in terms of vaccination parameters. The current distinction between the optimistic vs pessimistic estimates of vaccine efficacy is too limited. Specifically, the setting $VE_I = 0$ appears to be too limiting, and varying it within some bounds could be a subject of the sensitivity analysis. The effect of non-zero VE_I on mass vaccination can be high, and I would suggest to compare with the analysis and the results presented in Zachreson et al. (2021; <https://arxiv.org/abs/2103.07061>) - for obvious reasons, I do not suggest a citation, but I am interested in the Authors opinion of the role played by VE_I in the overall decomposition of the vaccine efficacy into its components, as well as its role in establishing robustness of the model.

In commenting on the model parameters, I assume that the latent period (specified as between 2 and 5 days, Table S3) matches the full incubation period, and the infectious period is related to the serial or generation period - this needs to be clarified. In particular, if the infectious period (specified as between 5.3 and 8.2 days) is meant to last beyond the serial/generation period, then it appears a little too short, and this needs to be discussed. As a possibly related issue, I note that R_0 is derived as 2.2, which again appears as too low. It may well be that a low R_0 is compensated by the shorter infectious period, and so the two lower estimates cancel each other out. If so, this can "backfire" in predicting epidemic dynamics, and may need to be explained or discussed. Another concern is the lack of consideration of symptomatic vs asymptomatic transmission. The argument that "asymptomatic persons ... represent a small fraction of the population at any given time" (lines 308-309) is dubious.

I also find the vaccination coverage level of 80% to be fairly unrealistic during the considered time horizon. Ideally, this level should also become a parameter investigated by the sensitivity analysis,

as it can clearly vary with age not only in terms of the timing when a fixed level (80%) is reached, but also in terms of the level itself. I believe that studying feasible age-dependent levels of vaccination coverage would be a valuable addition, and can easily leverage the strengths of the model which is already age-structured.

The paper argues that there is a strong agreement with the ECDC vaccination rollout rates (lines 130-131, "The ECDC vaccination rollout data for Portugal agree well with these projections"). Given the relatively early rollout stage, this may be too early to call, as Fig. 5b does not present a strong case. In any case, a revised submission may include a longer period of the vaccination rollout, putting these doubts to rest, or re-calibrating some aspects of the model. Incidentally, the paper argues that many other studies of mass-vaccination did not consider progressive rollout (lines 52-53: "these models typically assumed that a large proportion of the population is vaccinated instantaneously and/or did not focus on relaxation strategies"). While I agree that this is the case in general, making this submission a valuable addition to the literature, I note that ref. [45], which is grouped together with the pre-pandemic vaccination studies, did in fact consider a progressive rollout.

The manuscript states that "In the main analyses (Figures 6 and 7), the contact rates for vaccinated persons were equal to those unvaccinated, $c_{vk}(t) = c_{kl}(t)$ ". This would mean that there is no difference between equations (3) and (4), and the resultant forces of infection λ are the same for vaccinated and unvaccinated individuals. It would benefit the reader to discuss if this has any significance, especially in the context of the relaxation scenarios. Returning briefly to the role of VE_I in the model, where the only considered setting $VE_I = 0$ is used, equations (3) and (4) simplify, and again, it may be helpful to note and discuss this.

Subject to the limitations noted above, I find the results very convincing. The identified differences across considered scenarios are compelling and present actionable information to policy-makers. I would, nevertheless, suggest that the differences between scenario 4, on the one hand, and the scenarios 2, and especially 3, on the other hand, are distinguished more clearly. The natural protection achieved with scenario 3 is still $> 30\%$, and so how much better is scenario 4 needs to be articulated with respect to other indicators in addition.

This raises another point - the sensitivity analysis in terms of the model outcomes (rather than sensitivity/robustness of the model itself). Specifically, it needs to be quantified how the difference between scenarios 4 and 3 (and possibly, scenarios 4 and 2) changes when the vaccination coverage changes, VE_I changes, etc.

There are likely nonlinear effects in combining different intervention strategies, as has been reported in other epidemiological contexts. And so, it might be interesting to explore and discuss during which period and under which scenario do these combined effects appear stronger. This may present an opportunity to focus attention to specific factors affecting the relaxation scenarios.

A sentence around line 187 mentions "several orders of magnitude larger than the three previous waves", while the presented results seem to suggest only an order of magnitude larger. I may have misread the figures, but it would be nice to quantify the difference better.

A comment around lines 212-213 notes that "specific relaxation policies... are notoriously hard to implement in mechanistic transmission models". This comment is unclear if not questionable, as in fact, mechanistic agent-based models are quite efficient and effective in implementing various complex scenarios. For example, these models adequately account for non-heterogeneous home/work/study travel patterns - an aspect missing in the reviewed study. A comparison across model classes is not in scope for this work, and so comments on "mechanistic transmission models" are simply misplaced and unhelpful.

Overall, the paper is well-motivated, well-written, and is focussed on a well-defined case study,

approached at a sound technical level. I have no doubt that it would provide a valuable (and timely!) contribution to our efforts in controlling the COVID-19 pandemic around the world. It would be useful to articulate the general applicability of the outcomes, leveraging the case study, and discuss several international ramifications. In particular, some discussion of P.1 variant of concern (VoC) that emerged and spread in Brazil (which has well-established connections with Portugal), would be useful. While it is mentioned (ref. [30]), the impact of possible importation and spread of this VoC (more transmissible and exhibiting a higher immune escape) into Portugal during 2021 may need to be evaluated, or at least discussed in more detail.

Minor typos:

136 mid-march 2021 (March)

419 liner combination (linear)

Mikhail Prokopenko

Reviewer #3:

Remarks to the Author:

This article presents an excellent modelling work of the ongoing COVID-19 Epidemic in Portugal. The authors considered a compartmental SEIHR model at the national level. The model has been calibrated on data of hospitalizations and serological data through a Bayesian approach. The calibrated model has then been used to produce scenarios of the unfolding epidemic during Spring and Summer 2021, based on the possible lifting of the actual containment measures (and thus increasing the daily contacts among individuals) and the vaccination campaign. The results clearly show that the vaccination campaign is not sufficient to limit the spreading of the disease. The control of the transmission is obtained only through the gradual relaxation of the containment measures, together with the continuation of the vaccination campaigns.

The manuscript is well written, the presentation of the methods and results is clear. The results are highly of interest.

In the past year there have been many attempts to model COVID 19 epidemic at countries level, but only few works had this quality, presenting a detailed mathematical model, supported by epidemiological data and robust calibration.

I really enjoyed reading this paper: my opinion is that the authors did an excellent work.

Damiano Pasetto

Please find in the following few minor remarks and typos.

Figure 1 – caption: please note that the hospitalizations are grouped for different age classes then the one listed at the end of the caption.

Line 109 and Table 1: is there a reference for this data?

Figure S2 and Table S2: In table S2 the parameter beta is described as the ratio between the susceptibility of a age classes with respect to the susceptibility in age class n=10. Figure S2 shows beta for classes 0-20 and 20-60. I think that beta for age classes 60-80 are missing. I suggest moving Table S1 closer to Figure S2.

Figure 6: please increase the font size of the titles and the legend.

Line 419: replace 'liner' with 'linear'.

REVIEWER COMMENTS

Reviewer #2 (Remarks to the Author):

Dear Mikhail,

thank you for acknowledging that our study is a valuable and timely contribution to the efforts in controlling the COVID-19 pandemic. We appreciate your thorough review and your comments on how the work could be improved and made more clear to the readers. Following your suggestions, we have added several important sensitivity analyses and additional explanations throughout the text. The present version of the manuscript owes much to your review and we hope you find it suitable for publication without delay.

The paper studies effects of mass vaccination against SARS-CoV-2 (using Portugal as a case study), and considers several possible relaxation scenarios, feasible under the circumstances. The study is based on an age-structured transmission model calibrated to age-specific seroprevalence and hospital admissions data, as well as the projected vaccination coverage for Portugal. The impact of each considered scenario is quantitatively evaluated with respect to projected hospital admissions and the resultant effective reproduction number. Importantly, the paper distinguishes the achievable levels of the population immunization arising due to either natural infection or vaccination. The comparative analysis allows the Authors to quantify projected timing of controlling COVID-19 in Portugal.

The model used in this study has been partially validated previously, submission's ref. [43]. This work extends the model in several ways, most notably in considering effects of vaccination, by augmenting the model by several compartmental components accounting for vaccination (S^V , E^V , I^V , R^V and H^V - by the way, superscript V is missing in H^V in Fig. 8).

The typo in the model diagram has been rectified (updated Figure 8). Thank you for pointing it out.

The extended model retains the concept of cross-age contact interactions between the age groups, which is a strong point of the approach, making it much more refined than the traditional compartmental models, and increasing the resolution, while maintaining the power of analytical approaches. As such, it offers a clear way to combine the strengths of both compartmental models and higher-resolution agent-based models.

Nevertheless, as it is essentially a new model (not least, in terms of vaccination parameters), new sensitivity analysis needs to be performed and reported on. I was not able to find the outcomes of sensitivity analysis in the previous study (i.e., [43]), and some comments on the sensitivity of the model to changes in key epidemiological parameters, such as latent and infectious periods, would be beneficial. While the ranges of these parameters are provided and are within acceptable bounds, how robust is the model and its outcomes is unclear.

In the previous study (Reference 43), we presented the correlation matrix for the joint posterior density of the estimated parameters (Supplementary Figure 5) that reveals strong positive and negative correlations between some of the key epidemiological parameters (e.g. the infectious period, the initial fraction of infected individuals, the probability of transmission per contact and the hospitalization rate). In this respect, our model has not changed because the estimation procedure is very similar (Lines 467-470). However, the outcomes of the model such as the cumulative number of hospitalizations during the study period are not sensitive to the key epidemiological parameters among which the infectious period, the latent period and the probability of transmission per contact (Lines 470-474). See scatter plots in new Figure S12 made for Scenario 4 and a pessimistic set of vaccine efficacies where Pearson correlation coefficients between the three parameters and cumulative hospitalizations from 25 February 2020 till 24 June 2022 are in the range of 0.09 to 0.14.

Even more importantly, the sensitivity analysis needs to be performed in terms of vaccination parameters. The current distinction between the optimistic vs pessimistic estimates of vaccine efficacy is too limited. Specifically, the setting $VE_I = 0$ appears to be too limiting, and varying it within some bounds could be a subject of the sensitivity analysis. The effect of non-zero VE_I on mass vaccination can be high, and I would suggest to compare with the analysis and the results presented in

Zachreson et al. (2021; <https://arxiv.org/abs/2103.07061>) - for obvious reasons, I do not suggest a citation, but I am interested in the Authors opinion of the role played by VE_I in the overall decomposition of the vaccine efficacy into its components, as well as its role in establishing robustness of the model.

Thank you for the reference, we have cited it in the manuscript (new Reference 50: Zachreson C, Chang SL, Cliff OM, Prokopenko M. How will mass-vaccination change COVID-19 lockdown requirements in Australia?; arXiv 2103.07061, 2021.). We had not been aware of this study as prior to submission we searched for relevant preprints on medRxiv and published articles but we did not include arXiv in the search. The results presented in Zachreson et al. illustrate the role played by VE_I on the effects of mass vaccination. Because VE_I is unconstrained by the clinical efficacy VE_c , the authors explore the full range $[0,1]$ of this parameter when investigating, for instance, herd immunity requirements while the efficacy for susceptibility VEs and efficacy for disease VE_d are constrained by

$$VE_c = VE_s + VE_d - VE_s VE_d$$

and VE_c is taken as 0.9 for the priority vaccine and 0.6 for the general vaccine. In the notation of our manuscript, this is equation (14)

$$VE_{disease} = VE_{infection} + (1 - VE_{infection})VE_{disease | infection}$$

but we considered the ranges for $VE_{disease}$ (VE_c) and also for $VE_{infection}$ (VEs) reported in the literature and we assumed $VE_I = VE_{disease | infection}$. In particular, we took for the optimistic case $VE_{disease} = VE_{infection} = 94\%$ (and so $VE_I = VE_{disease | infection} = 0\%$). According to the results of Zachreson et al., and also according to intuition, changing the value of VE_I in a highly effective vaccine should have almost no effect. This is confirmed by our additional sensitivity analysis (new Figure S11 and Table S6; Lines 121-124, 203-209, 597-601, 605-606). For the pessimistic set we used $VE_{disease} = VE_{infection} = 55\%$ (and so again $VE_I = VE_{disease | infection} = 0\%$). For this lower efficacy, changing VE_I can be important. However, assuming $VE_I=0$ must correspond to the worst-case scenario, given any choice of the two constrained parameters. This is also confirmed by our additional sensitivity analysis (new Figure S11 and Table S6; Lines 121-124, 203-209, 597-601, 605-606).

Regarding the interplay of VEs and VE_d for a given VE_c , Zachreson et al. took in most their analysis $VE_s = VE_d$, while we took, following the literature and in particular [27], $VE_c = VE_s$ and $VE_d=0$. When combined with $VE_I=0$, the latter scenario is more favourable than the former. Still, according to the analysis also considered in Zachreson et al., Figure S1, we note that for most values of VE_I there is little difference between the two scenarios.

Summing up, the sensitivity analysis we carried on covers all possible scenarios for the overall decomposition of the vaccine efficacy into its components, with the exception of low VEs combined with low VE_I .

In commenting on the model parameters, I assume that the latent period (specified as between 2 and 5 days, Table S3) matches the full incubation period, and the infectious period is related to the serial or generation period - this needs to be clarified. In particular, if the infectious period (specified as between 5.3 and 8.2 days) is meant to last beyond the serial/generation period, then it appears a little too short, and this needs to be discussed.

We would like to point out that the key epidemiological parameters in our model, such as the latent ($1/\alpha$) and infectious ($1/\gamma$) periods, are not fixed but rather estimated in the Bayesian framework. Table S4 indicates prior distributions of these parameters and Figures S1 and S2 show the posterior distributions as estimated from the model fit to the hospitalization and seroprevalence data. The Bayesian framework implies that the relevant prior knowledge on the parameters may be incorporated in the estimation procedure and for the latent and infectious periods we used informative priors. We have adapted the captions to Figures S1 and S2 to stress it again.

Our choice of the range for the prior for the latent period (time between infection and becoming infectious) was based on the estimates of 4-6 days for the incubation period (time between infection and developing symptoms) (see newly added References 71 and 76: Park M et al. A Systematic Review of COVID-19 Epidemiology Based on Current Evidence. J Clin Med. 2020;9(4):967. doi:

10.3390/jcm9040967; Lauer SA et al. The Incubation Period of Coronavirus Disease 2019 (COVID-19) From Publicly Reported Confirmed Cases: Estimation and Application. *Ann Intern Med.* 2020;172(9):577-582. doi: 10.7326/M20-0504.). Since the current evidence suggests that SARS-CoV-2 transmission is possible before the development of symptoms, the latent period was chosen to be shorter than the incubation period, i.e. we used a narrower range of 2 to 5 days for 99% of the prior density of the latent period. We have added this explanation (Lines 451-456) and new references to Table S4 for the latent period.

We have now added a clarification (Lines 362-366) that in our previous model, we split the infectious period into $m = 3$ stages, thereby obtaining a more realistic distribution of the infectious period (i.e. Erlang/Gamma with shape parameter $m = 3$ instead of an Exponential distribution). For computational reasons (much longer time series and more complex projections) we could not implement an Erlang-distributed infectious period this time. The average generation interval in the Erlang/Gamma model is $1/\alpha + (1 + m)/(2*m)*1/\gamma$, where m is the shape parameter. In the Exponential model, $m = 1$ and the average generation interval is $1/\alpha + 1/\gamma$. We have now clarified in the text (Lines 456-463) that the priors on α and γ were chosen to match observed generation intervals (more precisely the serial interval: new Reference 77: Ali ST et al. Serial interval of SARS-CoV-2 was shortened over time by nonpharmaceutical interventions. *Science.* 2020;369(6507):1106–1109. doi:10.1126/science.abc9004.) of on average 7-8 days. Also note that the "effective" infectious period has likely decreased because (self-)isolation upon the development of respiratory symptoms has been recommended and, in certain situations, enforced (e.g. at schools, hospitals) during the course of the pandemic (Reference 77). Contact tracing and testing of asymptomatic persons also decreases the time of infectiousness. The mean a priori generation interval was 7.3 days (99%CrI 5.5-9.3). The individual serial intervals and duration of the infection have a much wider distribution. Table S4 contained a typo, as the 99% mass of the infectious period is (2.9-5.2) days. This is now corrected.

As a possibly related issue, I note that R_0 is derived as 2.2, which again appears as too low. It may well be that a low R_0 is compensated by the shorter infectious period, and so the two lower estimates cancel each other out. If so, this can "backfire" in predicting epidemic dynamics, and may need to be explained or discussed.

The estimated R_0 is indeed 2.2 but this is within the range of acceptable values. See e.g. new Reference 71: Park M et al. A Systematic Review of COVID-19 Epidemiology Based on Current Evidence. *J Clin Med.* 2020;9(4):967. doi: 10.3390/jcm9040967, where of 21 estimates for the basic reproduction number ranging from 1.9 to 6.5, 13 were between 2.0 and 3.0 and several studies estimated R_0 at 2.2 or lower. Moreover, Portugal is known for a relatively mild first pandemic wave which agrees with our work for the Netherlands (Ref 43), where R_0 for the Dutch pandemic was estimated at 2.71. The reason why R_0 estimate could be low for Portugal is seasonality in transmission for which our model does not account. We have now added a discussion on this (Lines 304-312):

Lastly, our analyses assume a casual relation between the control measures and the reduction in circulation of SARS-CoV-2, and do not incorporate seasonal variation in transmissibility. The data on human coronaviruses (229E, HKU1, NL63, OC43) from other locations (e.g. New York or Stockholm) show a marked seasonal pattern [Kissler2020,SaadRoy2020,Neher2020] with hardly any circulation in summer. If the seasonality played a major role in transmission of SARS-CoV-2 in summer 2020 then, irrespective of the relaxation of control measures, we could expect low epidemic activity near the end of spring 2021 too. The seasonality would also imply that the effective reproduction number is higher during the winter season and lower during the summer season, and could explain a relatively low basic reproduction number (median value of 2.20) estimated by the model in March 2020, although this value lies within the range of published estimates for other countries [Park2020].

Another concern is the lack of consideration of symptomatic vs asymptomatic transmission. The argument that "asymptomatic persons ... represent a small fraction of the population at any given time" (lines 308-309) is dubious.

There must have been misunderstanding regarding the model assumptions. Individuals enter the infectious compartment when they become infectious independently of whether they have symptoms or not and leave this compartment because they either get hospitalized (H) or recover without hospitalization (R). Therefore, this compartment contains both symptomatic and asymptomatic

individuals. There is absolutely no advantage in splitting individuals into symptomatic and asymptomatic in our model. For the contrary, we intentionally did not do it, because the parameters of such a model would not be identified from the model fit to the data streams we used. We have explained this in the model description (Lines 358-362).

The sentence the reviewer cited concerns the vaccination of infectious individuals. It was our choice not to include it because according to the current guidelines individuals have to be free from infection to be eligible for vaccination. We agree that in reality some fully asymptomatic individuals might be missed according to this definition, but even not vaccinating any infectious individuals at all has very little impact because the absolute majority of individuals at any given time are either in susceptible (S compartment) or in recovered states (H and R compartments) (next text on Lines 374-377).

I also find the vaccination coverage level of 80% to be fairly unrealistic during the considered time horizon. Ideally, this level should also become a parameter investigated by the sensitivity analysis, as it can clearly vary with age not only in terms of the timing when a fixed level (80%) is reached, but also in terms of the level itself. I believe that studying feasible age-dependent levels of vaccination coverage would be a valuable addition, and can easily leverage the strengths of the model which is already age-structured.

Portugal is a country where the overall intent to follow the advice of the government to get vaccinated is very high (Ref 58: Soares P et al. Factors Associated with COVID-19 Vaccine Hesitancy. *Vaccines*. 2021;9(3). doi:10.3390/vaccines9030300) which is not necessarily the case in other developed countries. Our choice for the vaccination coverage is based on the longitudinal survey data from this study for Portugal covering the period from 19 September 2020 till 5 February 2021 during which fewer than 5% of people indicated that they do not want to take the vaccine (attached presentation in Portuguese by our colleagues from the National School of Public Health for the Ministry of Public Health, page 7).

We agree, however, that the situation might change during the vaccination programme. In particular, as vaccination coverage in older age groups will start to saturate, younger people might have lower intent to get vaccinated. This is seen in Israel where it has proved to be difficult to reach high coverage in young people and in YouGov COVID-19 surveys from the UK (see Reference 27: Scientific Advisory Group for Emergencies. Imperial College London: Unlocking roadmap scenarios for England, 18 February 2021; 2021. Available from: <https://www.gov.uk/government/publications/imperial-college-london-unlocking-roadmap-scenarios-for-england-18-february-2021>). Based on the results from the UK survey we performed additional sensitivity analyses for the maximum vaccination coverage decreasing with age (90% in 80+ years old, 85% in 50 to 80-years-old, 75% in 20 to 50-years-old, and 0% in 0 to 20-years-old) (Figures S9 and S10, new text on Lines 124, 210-220, 608-610). The model predicts that in this case the cumulative median number of hospitalizations from 1 April 2021 till 1 January 2022 for Scenarios 1, 2 and 3 would be almost equal to the situation when the maximum vaccination coverage is independent of age and very high (Figure 5, 6 and 7, main text). For Scenario 4, the number of hospitalizations would be 8% higher than in Figure 7. The reason for this is that in Scenario 1 the pandemic unfolds much faster than vaccination is rolled out. In Scenarios 2 and 3 the pandemic is partially controlled

through the measures, i.e. the contact rates continue to be reduced till the end of 2021. In Scenario 4, though, the contact rates return to the pre-pandemic levels on 1 October 2021 while the coverage is not sufficiently high leading to a slight increase in hospitalizations at the end of 2021. This has been explained in the results section (Lines 210-220).

The paper argues that there is a strong agreement with the ECDC vaccination rollout rates (lines 130-131, "The ECDC vaccination rollout data for Portugal agree well with these projections"). Given the relatively early rollout stage, this may be too early to call, as Fig. 5b does not present a strong case. In any case, a revised submission may include a longer period of the vaccination rollout, putting these doubts to rest, or re-calibrating some aspects of the model.

The reviewer is correct in that the projections might change but so far there has been a really good agreement. The ECDC data file we used in the original manuscript has not been updated after the submission. Therefore, we have updated the data in Fig. 5b until 1 May 2021 based on the number of fully vaccinated persons by ourworldindata.org (Ref [1]). The coverage might start to increase faster in the second half of 2021 due to vaccine new contracts of the Portuguese government but this is also accounted for by the model (the slope of the coverage increases on 1 May 2021).

Incidentally, the paper argues that many other studies of mass-vaccination did not consider progressive rollout (lines 52-53: "these models typically assumed that a large proportion of the population is vaccinated instantaneously and/or did not focus on relaxation strategies"). While I agree that this is the case in general, making this submission a valuable addition to the literature, I note that ref. [45], which is grouped together with the pre-pandemic vaccination studies, did in fact consider a progressive rollout.

Reference 45, previous version of the manuscript: Bartsch et al. Vaccine Efficacy Needed for a COVID-19 Coronavirus Vaccine to Prevent or Stop an Epidemic as the Sole Intervention. *American Journal of Preventive Medicine*. 2020;59(4):493–503. doi:10.1016/j.amepre.2020.06.011. did NOT consider a progressive rollout. This paper states on page 502 subsection Limitations "Simulation experiments assumed an optimistic situation in which the population was vaccinated in 1 day. In reality, vaccination would take place over an extended period, which would reduce the impact of vaccination".

Limitations

All models, by definition, are simplifications of real life and cannot account for every possible outcome.²¹ Model inputs drew from various sources, and new data on SARS-CoV-2 continue to emerge. For example, estimates of R_0 vary widely and in many studies, are calculated considering only symptomatic infections.^{11–13} Because the course of an actual SARS-CoV-2 epidemic may not be very predictable, this study explored a range of possible scenarios and parameter values in sensitivity analyses. In addition, individuals mixed equally with each other, whereas in actuality, transmission and thus vaccination's impact may be higher among closer contacts, raising the possibility that targeted vaccination approaches may have a higher impact. Simulation experiments assumed an optimistic situation in which the population was vaccinated in 1 day. In reality, vaccination would take place over an extended period, which would reduce the impact of vaccination. Moreover, assumptions about levels of herd immunity required may depend on differing individual susceptibility to SARS-CoV-2. The model also assumes that there are sufficient healthcare resources (e.g., intensive care unit beds and ventilators) for all patients. However, if the healthcare system is overburdened, patients with COVID-19 may not receive proper care, leading to higher mortality.

If the Reviewer meant Reference 44, that study did not focus on the relaxation strategies as our text states (Lines 52-53).

The manuscript states that "In the main analyses (Figures 6 and 7), the contact rates for vaccinated persons were equal to those unvaccinated, $cVkl(t) = ckl(t)$ ". This would mean that there is no difference between equations (3) and (4), and the resultant forces of infection λ are the same for vaccinated and unvaccinated individuals. It would benefit the reader to discuss if this has any significance, especially in the context of the relaxation scenarios. Returning briefly to the role of VE_I in the model, where the only considered setting $VE_I = 0$ is used, equations (3) and (4) simplify, and again, it may be helpful to note and discuss this.

The reviewer is correct in interpreting our main analyses and we find this suggestion really useful. The results for the case when the contact rates of vaccinated and unvaccinated are not equal have been added in the Figures S7, S8 and Table S2 for Scenario 4 as the most realistic scenario for the future relaxation of measures (next text Lines 123-124, 192-209, 605-608). This is the scenario which we refer to as behavior compensation post-vaccination, i.e. individuals return to pre-pandemic contact rates immediately upon getting vaccinated. Both in the presence and in the absence of behavior compensation in Scenario 4, the number of breakthrough cases after vaccination is relatively small (about 5 to 6% of the total cumulative cases) for optimistic vaccine efficacies and is comparable to the number of infections in the unvaccinated population (about 42% to 46% of the total infections) for pessimistic vaccine efficacies (Figure S8 and Table S2). Overall, the change in behavior of vaccinated persons would have relatively little impact on cumulative hospital admissions, i.e. an increase of 5% from 4,088 to 4,301 for optimistic vaccine efficacies and an increase of 4% from 30,028 to 31,344 for pessimistic vaccine efficacies (Figure S7 and Table S2). Therefore, the model projections for hospital admissions depend more strongly on our assumptions regarding pessimistic and optimistic vaccine efficacies (i.e. several fold increase in hospitalizations for the range explored) and to a smaller extent on the assumptions regarding behavioral changes in the vaccinated population (i.e. few percent increase in hospitalizations for the range explored). We fully agree that the sensitivity with respect to VE_I needed to be explored and commented upon which, as mentioned above, we have now done in Figure S11, Table S6 and the text (Lines 121-124, 203-209, 597-601, 605-606).

Subject to the limitations noted above, I find the results very convincing. The identified differences across considered scenarios are compelling and present actionable information to policy-makers. I would, nevertheless, suggest that the differences between scenario 4, on the one hand, and the scenarios 2, and especially 3, on the other hand, are distinguished more clearly. The natural protection achieved with scenario 3 is still $> 30\%$, and so how much better is scenario 4 needs to be articulated with respect to other indicators in addition.

From the public health perspective, the most relevant outcome measure is hospital admissions. In all cases, the comparison was done and commented upon in the text based on the cumulative median number of hospitalizations after the start of the relaxation of measures until the end of the study period. Specifically, the number of hospitalizations in Scenario 4 is 2.2 times larger than in Scenario 3 (3,194 vs 1,450 from 1 April 2021 till 1 January 2022) and 2.8 times smaller than in Scenario 2 (3,194 vs 8,975 in the same time period) but Scenario 4 would still seem manageable for the healthcare system because the model does not predict sharp increases in hospital admissions (next text on Lines 168-174). Most importantly, unlike in Scenarios 2 and 3 where contact rates stay reduced after 1 April 2021, the return to pre-pandemic contact patterns in Scenario 4 is gradual and the complete lifting of measures occurs on 1 October 2021 which would have important socio-economic consequences. Since we use an epidemiological model, we do not further quantify the impact of the reduced contact rates and only focus on relevant infections leading to hospitalizations.

This raises another point - the sensitivity analysis in terms of the model outcomes (rather than sensitivity/robustness of the model itself). Specifically, it needs to be quantified how the difference between scenarios 4 and 3 (and possibly, scenarios 4 and 2) changes when the vaccination coverage changes, VE_I changes, etc.

A subsection was added in the Methods section to summarize the performed sensitivity analyses (Lines 602-612). The sensitivity analyses have been conducted with respect to (i) the maximum vaccination coverage decreasing with age for Scenarios 1, 2, 3, and 4, i.e. the coverage for ages [0,20), [20,50), [50,80), and 80+ was 0%, 75%, 85%, and 90%, respectively (new Figures S9 and

S10); (ii) behavior compensation post-vaccination using, as an example, Scenario 4, that was modelled as a return to pre-pandemic contact rates among vaccinated persons as compared to unvaccinated persons who may continue to have reduced contact rates due to control measures (new Figures S7, S8 and Table S2); (iii) optimistic and pessimistic set of vaccine efficacies in Scenario 4 (new Figure S7, S8 and Table S2); (iv) timings of different relaxation steps in Scenario 4 where Step 3 occurs on 1 August instead of 1 October 2021 (Figure S6).

There are likely nonlinear effects in combining different intervention strategies, as has been reported in other epidemiological contexts. And so, it might be interesting to explore and discuss during which period and under which scenario do these combined effects appear stronger. This may present an opportunity to focus attention to specific factors affecting the relaxation scenarios.

It is not clear what the reviewer meant by nonlinear effects and there is no specific reference supporting the requested change. Given that the discussion has already been extended in a number of important ways and currently occupies three pages, we have decided to focus it on the most relevant and pressing points only. In particular, we have added a comment on an important (in the authors' view) model limitation which is the absence of seasonality in transmission (Lines 304-312).

A sentence around line 187 mentions "several orders of magnitude larger than the three previous waves", while the presented results seem to suggest only an order of magnitude larger. I may have misread the figures, but it would be nice to quantify the difference better.

As requested, we have quantified the exact difference using the cumulative median number of hospitalizations and rephrased the sentence as follows: Even for the most optimistic model assumptions, this scenario would result in a wave of hospitalizations 20% larger than the three previous waves combined together (58,226 cumulative median hospitalizations from 1 April 2021 till 1 January 2022 versus 48,273 hospitalizations from 25 February 2020 till 31 March 2021) (Lines 228-231).

A comment around lines 212-213 notes that "specific relaxation policies... are notoriously hard to implement in mechanistic transmission models". This comment is unclear if not questionable, as in fact, mechanistic agent-based models are quite efficient and effective in implementing various complex scenarios. For example, these models adequately account for non-heterogeneous home/work/study travel patterns - an aspect missing in the reviewed study. A comparison across model classes is not in scope for this work, and so comments on "mechanistic transmission models" are simply misplaced and unhelpful.

Thank you for pointing out that this sentence was unclear. What we meant by "specific relaxation policies" are policies that are not straightforward to implement in a compartmental model like ours even if location-specific matrices were available. Examples of such relaxation measures are e.g. contact tracing (Reference 33: Kretzschmar ME, Rozhnova G et al. Impact of delays on effectiveness of contact tracing strategies for COVID-19: a modelling study. *The Lancet Public Health*. 2020;5(8):e452-e459. doi:10.1016/S2468-2667(20)30157-2) or other policies in which governments might be interested but that do not have exact interpretation in terms of contact matrices (e.g. "banned gatherings of more than 3 people and family members of patients have to stay at home"). We have clarified this in the discussion (Lines 256-261).

Overall, the paper is well-motivated, well-written, and is focussed on a well-defined case study, approached at a sound technical level. I have no doubt that it would provide a valuable (and timely!) contribution to our efforts in controlling the COVID-19 pandemic around the world. It would be useful to articulate the general applicability of the outcomes, leveraging the case study, and discuss several international ramifications. In particular, some discussion of P.1 variant of concern (VoC) that emerged and spread in Brazil (which has well-established connections with Portugal), would be useful. While it is mentioned (ref. [30]), the impact of possible importation and spread of this VoC (more transmissible and exhibiting a higher immune escape) into Portugal during 2021 may need to be evaluated, or at least discussed in more detail.

This is an interesting suggestion. The P.1 VoC does not appear to be on the rise in Portugal and, to our knowledge, there is no evidence that its transmissibility makes it more competitive than the B.1.1.7 VoC which is already dominant in Portugal. Specifically, the frequency of the P.1 lineage

(known as 20J/501Y.V3) in samples collected at the national level by the Portuguese reference laboratory (INSA) were: Jan 2021: 0% (n=532), Feb 2021: 0.4% (n=861), Mar 2021: 0.4% (n=1094) (see the attached report in Portuguese by INSA for the Ministry of Public Health).

Tabela 1. Frequência relativa das principais variantes genéticas do SARS-CoV-2 detectadas na amostragens nacionais de Janeiro, Fevereiro e Março de 2021*, bem como o número total de sequências dessas variantes detectadas até à data (n=5758). A Figura 2 apresenta a evolução da frequência relativa dessas variantes desde Novembro de 2020, mantendo a correspondência de cores apresentada nas Tabela 1 e 2.

Variante / linhagem	Frequência relativa na amostragem nacional, 2021			Total de sequências até à data (n = 5758)	CONTEXTO
	Janeiro (n=532)	Fevereiro (n=861)	Março (n= 1094)		
S:D614G+N501Y+H69del/V70del++ (clade 20J; VOC-202012/01; 501Y.V1; linhagem B.1.1.7)	16.0%	58.2%	82.9%	1744	- Esta variante, originalmente detetada no Reino Unido, está associada a uma maior capacidade de transmissão. https://www.ecdc.europa.eu/sites/default/files/documents/SARS-CoV-2-variant-multiple-spike-protein-mutations-United-Kingdom.pdf
S:D614G+A222V (clade "20E / EU1"; linhagem e sub-linhagens B.1.177)	54.7%	23.5%	8.3%	1336	- Este cluster (sub-clade 20A.EU1) terá tido origem em Espanha (em meados de Junho), tendo revelado uma elevada disseminação a nível global. Hodcroft et al, medRxiv. https://www.medrxiv.org/content/10.1101/2020.10.25.20219063v2
S:D614G+S477N (sub-clade "20A.EU2"; linhagem B.1.160)	13.2%	7.0%	2.5%	333	- A mutação S477N na Spike altera o domínio de ligação ao receptor ("RBD") ACE2, podendo aumentar a afinidade da ligação da Spike às células do hospedeiro. Starr et al, 2020, Cell. https://www.sciencedirect.com/science/article/pii/S0092867420310035
S:D614G+L452R++ (sub-clade 20D; linhagem C.16)	6.8%	5.1%	0.8%	114	- Esta variante apresenta a mutação L452R, a qual se prevê afectar o domínio de ligação da proteína Spike ao receptor ("RBD") ACE2, podendo mediar resistência a anticorpos neutralizantes. Li et al, 2020, Cell. https://www.sciencedirect.com/science/article/abs/pii/S0092867420308771
S:D614G+N439Y+H69del/V70del (sub-clade 20A; sub-linhagens B.1.258)	1.7%	0.1%	0.1%	23	- A mutação N439Y na Spike altera o domínio de ligação ao receptor ("RBD") ACE2, podendo aumentar a afinidade da ligação da Spike às células do hospedeiro e alterar o reconhecimento por anticorpos. Barnes et al, 2020, Nature https://www.nature.com/articles/s41586-020-2852-1
S:D614G+S98F (sub-clade 20A; linhagem B.1.221)	1.1%	0.5%	0.1%	39	- Esta variante apresentou maior frequência na Bélgica e Holanda, tendo já sido identificada em múltiplos países. Hodcroft et al, medRxiv. https://www.medrxiv.org/content/10.1101/2020.10.25.20219063v2 ; https://github.com/emmahodcroft/cluster_scripts/blob/master/README.md#sn501
S:D614G+E484K++ (clade 20B; linhagem P.2)	0.6%	0.6%	0.1%	18	Esta variante apresenta uma mutação no domínio de ligação ao receptor ("RBD") ACE2 da proteína Spike, a qual é potencialmente mediadora de resistência a anticorpos neutralizantes (S: E484K). Foi recentemente associada a casos de re-infecção no Brasil. Resende et al, 2021, virological. https://virological.org/t/spike-e484k-mutation-in-the-first-sars-cov-2-reinfection-case-confirmed-in-brazil-2020/584
S:D614G+N501Y+E484K++ (clade 20H; 501Y.V2; linhagem B.1.351)	0.0%	0.1%	2.5%	49	- Esta variante apresenta mutações na proteína Spike, potencialmente mediadoras de uma maior capacidade de transmissão (ex. S: N501Y) e de resistência a anticorpos neutralizantes (ex. S: E484K). Foi detectada pela primeira vez na África do Sul. Tegally et al, 2020, medRxiv. https://www.medrxiv.org/content/10.1101/2020.12.21.20248640v1
S:D614G+N501Y+E484K++ (clade 20J; 501Y.V3; linhagem P.1)	0%	0.4%	0.4%	22	- Esta variante apresenta múltiplas mutações na proteína Spike, potencialmente mediadoras de uma maior capacidade de transmissão (ex. N501Y) e/ou evasão ao sistema imunitário (ex. E484K). Faria et al 2021 https://virological.org/t/genomic-characterisation-of-an-emergent-sars-cov-2-lineage-in-manaus-preliminary-findings/586

Experimental studies have demonstrated that this variant may evade neutralizing antibody responses induced by infection and vaccination (e.g. newly added Reference 69: Hoffmann M et al. SARS-CoV-2 variants B.1.351 and P.1 escape from neutralizing antibodies. Cell. 2021 Mar 20;S0092-8674(21)00367-6. doi: 10.1016/j.cell.2021.03.036.). For this reason, the spread of this variant later during 2021 would be more compatible with model scenarios performed for lower vaccine efficacies which we mentioned in the discussion (Lines 299-303).

Minor typos:

136 mid-march 2021 (March)
419 liner combination (linear)

We have corrected the typos.

Kind regards,
Ganna Rozhnova
on behalf of all co-authors

Mikhail Prokopenko

Reviewer #3 (Remarks to the Author):

This article presents an excellent modelling work of the ongoing COVID-19 Epidemic in Portugal. The authors considered a compartmental SEIHR model at the national level. The model has been calibrated on data of hospitalizations and serological data through a Bayesian approach. The calibrated model has then been used to produce scenarios of the unfolding epidemic during Spring and Summer 2021, based on the possible lifting of the actual containment measures (and thus increasing the daily contacts among individuals) and the vaccination campaign. The results clearly show that the vaccination campaign is not sufficient to limit the spreading of the disease. The control of the transmission is obtained only through the gradual relaxation of the containment measures, together with the continuation of the vaccination campaigns.

The manuscript is well written, the presentation of the methods and results is clear. The results are highly of interest.

In the past year there have been many attempts to model COVID 19 epidemic at countries level, but only few works had this quality, presenting a detailed mathematical model, supported by epidemiological data and robust calibration.

I really enjoyed reading this paper: my opinion is that the authors did an excellent work.

Damiano Pasetto

Dear Damiano,

thank you for the careful reading of our manuscript and recognizing the quality of our study. We have addressed your suggestions below. We have also improved the manuscript by adding a number of important sensitivity analyses requested by the second reviewer. We hope that you find the current version of the manuscript suitable for publication.

Please find in the following few minor remarks and typos.

Figure 1 – caption: please note that the hospitalizations are grouped for different age classes then the one listed at the end of the caption.

We have modified the last two sentences of the caption of Figure 1 to clarify that

Hospital admissions were estimated for 10 age groups (see Methods). For presentation purposes, we grouped hospitalizations for ages [0,5), [5,10), [10,20) into the group of [0,20), so only 8 age groups are shown in this figure.

Line 109 and Table 1: is there a reference for this data?

Co-author Manuel Gomes who serves as Member of the National Immunization Technical Advisory Group against COVID-19 in Portugal was one of the developers of this Table/Program. It is shared in its original form on the GitHub for this manuscript [<https://github.com/lynxgav/COVID19-vaccination>] as VaccinationPlan.xlsx. We have included the reference to the GitHub directly in the text (Line 110).

Figure S2 and Table S2: In table S2 the parameter beta is described as the ratio between the susceptibility of a age classes with respect to the susceptibility in age class n=10. Figure S2 shows beta for classes 0-20 and 20-60. I think that beta for age classes 60-80 are missing. I suggest moving Table S1 closer to Figure S2.

The susceptibility of 60+ age group was used as the reference, i.e. beta for 60+ = 1. We have now added this sentence to the caption of Figure S2 and to Table S1 (former Table S2) and positioned them next to each other.

Figure 6: please increase the font size of the titles and the legend.

We know the figure editing to adapt fonts types and sizes will be requested by the journal at a later stage. We prefer to do it after the manuscript has been finalized.

Line 419: replace 'liner' with 'linear'.

The typo has been corrected.

Kind regards,
Ganna Rozhnova
on behalf of all co-authors

Reviewers' Comments:

Reviewer #2:

Remarks to the Author:

Thank you for a comprehensive revision, which improved the paper in several ways. In particular, the new sensitivity analyses include now the case with the vaccine transmission efficacy $V_I = 0.5$. Importance of this case is further emphasized by the recent study of Harris et al., which confirmed that the efficacy against transmission for BNT162b2 is close to this level: "Impact of vaccination on household transmission of SARS-COV-2 in England", preprint, 2021; <https://khub.net/documents/135939561/390853656/Impact+of+vaccination+on+household+transmission+of+SARS-COV-2+in+England.pdf/35bf4bb1-6ade-d3eb-a39e-9c9b25a8122a?t=1619601878136> . In other words, the case with $V_I = 0.5$ is a realistic case.

The added explanation of the model parameters is adequate. The low estimate for the reproduction number R_0 (2.2) is somewhat explained by the conjectured seasonality aspect. However, in light of the variants of concerns spreading in Portugal, such as B.1.1.7 which has a higher transmissibility, a more adequate estimate for R_0 would be much higher. For example, a study by Davies et al. ("Estimated transmissibility and impact of SARS-CoV-2 lineage B.1.1.7 in England", *Science*, 2021) estimated a 77% (95% CI, 73 to 81%) increase in the reproduction number R for UK; as well as 55% (95% CI, 45 to 66%) higher in Denmark, 74% (95% CI, 66 to 82%) higher in Switzerland, and 59% (95% CI, 56 to 63%) higher in the United States. Their alternative estimation approach also yielded similar increases between 43% and 57%.

And so I still find that the level of R_0 adopted in the reviewed submission (which aimed to trace epidemic scenarios for 2021) is too low. I understand that the model is calibrated to actual data, and the relatively low R_0 is compensated somehow by the choice of other parameters, but this needs to be mentioned among the limitations (at the very least). While $R_0 = 2.2$ is within the range of published estimates for other countries (for 2020), this value is out of range for 2021 in countries where B.1.1.7 is dominant (even the lowest estimate of 1.9 cited by the Authors [Park2020] would significantly increase beyond 2.2 when the higher transmissibility is accounted for).

A minor point on the study of Bartsch et al. ("Vaccine Efficacy Needed for a COVID-19 Coronavirus Vaccine to Prevent or Stop an Epidemic as the Sole Intervention. *American Journal of Preventive Medicine*. 2020;59(4):493–503). This work did differentiate between a pre-pandemic intervention and a vaccination campaign during an ongoing epidemic, specifically quantifying the outcomes of scenarios when vaccination occurs after 5%, 15%, and 30% of the population has already been exposed. I agree, however, that this comparative analysis is not the same as modelling a "progressive rollout". It would be helpful, nevertheless, if the difference is articulated better, highlighting that the model under consideration overcomes the limitation of instantaneous interventions.

I also agree that exploring nonlinear effects in combining different intervention strategies may not be needed in this paper. A recent report by Borchering et al. ("Modeling of Future COVID-19 Cases, Hospitalizations, and Deaths, by Vaccination Rates and Nonpharmaceutical Intervention Scenarios — United States, April–September 2021"; <https://www.cdc.gov/mmwr/volumes/70/wr/mm7019e3.htm>) is a good example, illustrating some of the nonlinearity I mentioned. In particular, Fig. 1 shows how different levels of vaccination nonlinearly combine with the NPI levels. A comment on such nonlinear effects may be useful.

A minor typo in Fig. 8: H_V should be H^V :)

In summary, I think the paper can be accepted, with the only caveat being a clarification of the

low R0 (in light of the findings of Davies et al.), and possible additions related to studies of Harris et al., and Borchering et al. (I have no relation to any of these groups).

REVIEWERS' COMMENTS

Reviewer #2 (Remarks to the Author):

Dear Mikhail,

thank you for providing comments on the revised manuscript. We believe there was a huge misunderstanding in how we modelled the spread of the British variant in Portugal. We explain it in more detail and address other suggestions below.

Thank you for a comprehensive revision, which improved the paper in several ways. In particular, the new sensitivity analyses include now the case with the vaccine transmission efficacy $V_I = 0.5$. Importance of this case is further emphasized by the recent study of Harris et al., which confirmed that the efficacy against transmission of BNT162b2 is close to this level: "Impact of vaccination on household transmission of SARS-COV-2 in England", preprint, 2021; <https://khub.net/documents/135939561/390853656/Impact+of+vaccination+on+household+transmission+of+SARS-COV-2+in+England.pdf/35bf4bb1-6ade-d3eb-a39e-9c9b25a8122a?t=1619601878136>. In other words, the case with $V_I = 0.5$ is a realistic case.

The added explanation of the model parameters is adequate. The low estimate for the reproduction number R_0 (2.2) is somewhat explained by the conjectured seasonality aspect. However, in light of the variants of concerns spreading in Portugal, such as B.1.1.7 which has a higher transmissibility, a more adequate estimate for R_0 would be much higher. For example, a study by Davies et al. ("Estimated transmissibility and impact of SARS-CoV-2 lineage B.1.1.7 in England", Science, 2021) estimated a 77% (95% CI, 73 to 81%) increase in the reproduction number R for UK; as well as 55% (95% CI, 45 to 66%) higher in Denmark, 74% (95% CI, 66 to 82%) higher in Switzerland, and 59% (95% CI, 56 to 63%) higher in the United States. Their alternative estimation approach also yielded similar increases between 43% and 57%.

And so I still find that the level of R_0 adopted in the reviewed submission (which aimed to trace epidemic scenarios for 2021) is too low. I understand that the model is calibrated to actual data, and the relatively low R_0 is compensated somehow by the choice of other parameters, but this needs to be mentioned among the limitations (at the very least). While $R_0 = 2.2$ is within the range of published estimates for other countries (for 2020), this value is out of range for 2021 in countries where B.1.1.7 is dominant (even the lowest estimate of 1.9 cited by the Authors [Park2020] would significantly increase beyond 2.2 when the higher transmissibility is accounted for).

We finally understand the Reviewer's concern regarding the value of R_0 and the increased transmissibility of the B.1.1.7 variant. In fact, our original submission already accounted for the increased transmissibility of the British variant in 2021 (and further). Lines 426-430 (current version but the same text was included in the original submission):

The rapid spread of B.1.1.7 variant, that is estimated to be about 50% more transmissible based on the data from England (Davies2021a, Volz2020, Graham2021), fueled the third wave of hospitalizations in Portugal. The increasing dominance of this variant was modelled empirically as a gradual increase in the probably of transmission per contact by 50% as follows $\epsilon * [1 + 0.5 / (1 + \exp(-K_0(t - t_{data})))]$, where ϵ and K_0 were estimated based on the data until 15 January 2021 (Supplementary Figure 2) and t_{data} is the last date in the hospital admission data (15 January 2021).

What we do is increasing the probability of transmission per contact by 50% in the beginning of 2021 after it is estimated based on the data from February 2020 till January 2021. Therefore, the R_e values in 2021 are consequently increased by 50%. This however does not mean that R_0 has to be 50% higher because R_0 by definition was computed in the time period when the pandemic just started (February 2020). Given that we do exactly what the Reviewer requests, we think this fully addresses the Reviewer's concern. To avoid any misunderstanding, we added a note to Supplementary Table 1 to explain that ϵ is increased by 50% in the beginning of 2021.

A minor point on the study of Bartsch et al. ("Vaccine Efficacy Needed for a COVID-19 Coronavirus Vaccine to Prevent or Stop an Epidemic as the Sole Intervention. American Journal of Preventive

Medicine. 2020;59(4):493–503). This work did differentiate between a pre-pandemic intervention and a vaccination campaign during an ongoing epidemic, specifically quantifying the outcomes of scenarios when vaccination occurs after 5%, 15%, and 30% of the population has already been exposed. I agree, however, that this comparative analysis is not the same as modelling a "progressive rollout". It would be helpful, nevertheless, if the difference is articulated better, highlighting that the model under consideration overcomes the limitation of instantaneous interventions.

This reference appears in the context of the comparison to our own study. Specifically, we wrote that the models in (Bubar2021,Bartsch2020,Makhoul2020,Matrajt2020,Moore2020) assumed that a large proportion of the population is vaccinated instantaneously or did not focus on relaxation strategies. The fact that the study by Bartsch et al considered different initial fraction of people who have been infected with the virus does not mean that they look at the progressive rollout of vaccination (e.g. a realistic plan of how the vaccination coverage will be growing with time) and how the measures would need to be relaxed (at the same time). As we wrote before, in this model all population is vaccinated within 1 day. Moreover, the study by Bartsch et al is not the only one that considered different initial conditions. We would like to point out that in introduction we purposefully want to mention what we do differently instead of going into details and assumptions of all other studies.

I also agree that exploring nonlinear effects in combining different intervention strategies may not be needed in this paper. A recent report by Borchering et al. ("Modeling of Future COVID-19 Cases, Hospitalizations, and Deaths, by Vaccination Rates and Nonpharmaceutical Intervention Scenarios — United States, April–September 2021"; <https://www.cdc.gov/mmwr/volumes/70/wr/mm7019e3.htm>) is a good example, illustrating some of the nonlinearity I mentioned. In particular, Fig. 1 shows how different levels of vaccination nonlinearly combine with the NPI levels. A comment on such nonlinear effects may be useful.

Thank you for this reference. As we explained in our previous report, it is our choice to limit the discussion to the most important points only. Currently the discussion occupies two full pages, the manuscript by far exceeds 5000 words limit and will have to be shortened. Moreover, it already contains 79 references (instead of the recommended 70 references). We have aimed to include different groups and be representative in our choice of references but it is also not realistic to include more and to focus on all non-essential discussions.

A minor typo in Fig. 8: H_V should be H^V :)

We introduced another typo during the first revision that has been corrected in the second revision ($\$H_V\$$ to $\$H^V\$$).

In summary, I think the paper can be accepted, with the only caveat being a clarification of the low R_0 (in light of the findings of Davies et al.), and possible additions related to studies of Harris et al., and Borchering et al. (I have no relation to any of these groups).

Thank you. As we explained above, the reference list is already too long (79 instead of 70 references) so we do not think it is not possible to add more otherwise they will anyway have to be removed at a later stage.

Mikhail Prokopenko

Kind regards,
Ganna Rozhnova